

# The oceanic cycle of carbon monoxide and its emissions to the atmosphere

Ludivine Conte[1], Sophie Szopa[1], Roland Séférian[2], Laurent Bopp[3]

[1]Laboratoire des Sciences du Climat et de l'Environnement, CEA/CNRS/UVSQ, Gif sur Yvette, 91191, France
[2]Centre National de Recherches Météorologiques, Météo-France/CNRS, Toulouse, 31057, France
[3]Laboratoire de Météorologie Dynamique, ENS, Paris, 75005, France

*Correspondence to*: Ludivine Conte (ludivine.conte@lsce.ipsl.fr)

**Abstract.** The ocean is a source of atmospheric carbon monoxide (CO), a key component for the oxidizing capacity of the atmosphere. It constitutes a minor source at the global scale, but could play an important role far from continental anthropized emission zones. To date, this natural source is estimated with large uncertainties, especially because the processes driving the oceanic CO are related to the biological productivity and can thus have a large spatial and temporal variability. Here we use the NEMO-PISCES (Nucleus for European Modelling of the Ocean, Pelagic Interaction Scheme for Carbon and Ecosystem Studies) ocean general circulation and biogeochemistry model to dynamically assess the oceanic CO budget and its emission to the atmosphere at the global scale. The main bio-chemical sources and sinks of oceanic CO are explicitly represented in the model. The sensitivity to different parameterizations is assessed. In combination to the model, we present here the first compilation of literature reported *in situ* oceanic CO data, collected around the world during the last 50 years. The main processes driving the CO concentration are photoproduction and bacterial consumption and are estimated to 19.2 and 21.9 Tg C yr$^{-1}$ respectively with our best-guess modelling setup. There are however very large uncertainties on their respective magnitude. Despite the scarcity of the *in situ* CO measurements in terms of spatio-temporal coverage, the proposed best simulation is able to represent most of the data (~300 points) within a factor of two. Overall, the global emissions of CO to the atmosphere are 3.6 Tg C yr$^{-1}$, in the range of recent estimates, but very different from the ones published by Erickson in 1989, which were the only gridded global emission available to date. These oceanic CO emission maps are relevant for use by atmospheric chemical models, especially to study the oxidizing capacity of the atmosphere above the remote ocean.

## 1 Introduction

Atmospheric carbon monoxide (CO) plays an important role in atmospheric chemistry. It indirectly affects the lifetime of greenhouse gases like methane (CH$_4$) as it is the dominant sink for tropospheric hydroxyl radicals (Thompson, 1992; Taylor et al. 1996), and impacts air quality as it is involved in ozone chemistry (Crutzen, 1974; Cicerone, 1988). With the increasing concern about atmospheric pollution and the potential role of CO, one of the first motivation to study the oceanic CO concentrations was to evaluate the relative stability of its trend in the marine boundary layer (Swinnerton and



Lamontagne, 1973). In 1968, Swinnerton et al. conducted in the Mediterranean Sea the first measurements of oceanic CO and reported a supersaturation of a few orders of magnitude in the surface waters with respect to the partial pressure of this gas in the atmosphere. Subsequent cruises confirmed the supersaturation of oceanic CO concentrations (Seiler and Junge, 1970; Swinnerton et al., 1970; Lamontagne et al., 1971; Swinnerton and Lamontagne, 1973; Linnenbom et al., 1973) and

supported the idea that the world oceans serve as a source of CO for the atmosphere.

The CO concentration in the surface ocean is of a few nmol L$^{-1}$ and presents large diurnal variations with a characteristic minimum just before dawn and a maximum in the early afternoon (Swinnerton et al., 1970; Bullister et al., 1982; Conrad et al., 1982). Rapid changes in the oceanic concentration result from the interplay between strong source and sink processes and are associated to a small CO residence time of a few hours to a few days (Jones, 1991; Johnson and

Bates, 1996). The photolysis of colored dissolved organic matter (CDOM) is thought to be the main source of CO in the ocean (Wilson et al., 1970; Redden, 1982; Gammon and Kelly, 1990; Zuo and Jones, 1995). CDOM is the photo-absorbing part of the dissolved organic matter (DOM) pool. Its absorption of solar radiations (mainly in the ultraviolet and blue wavelengths) initiates oxidation reactions that lead to the formation of a number of stable compounds, mainly $CO_2$ and CO (Miller and Zepp, 1995; Mopper and Kieber, 2001). Based on surface CDOM estimates from remote sensing and a model of

CO production, Fichot et Miller (2010) have estimated this source as 41 Tg C yr$^{-1}$. Additional sources of CO have also been reported. Especially, direct production by phytoplankton has been observed in laboratory experiments (Gros et al., 2009). However, production pathways remain unclear and this source is still considered to be a minor contributor for the global ocean compared to marine photochemistry (Tran et al., 2013; Fichot and Miller, 2010). The main sinks for oceanic CO are the microbial uptake (Seiler, 1978; Conrad and Seiler, 1980; Conrad et al. 1982) and the sea-to-air fluxes (Jones, 1991;

Doney et al., 1995). Diverse communities of marine bacteria have been shown to oxidize CO (Tolli et al., 2006; King and Weber, 2007), mainly for a supplemental source of energy (Moran and Miller 2007). Rates and kinetics of this biological sink are not well known, even if this sink may be responsible for about 86% of the global oceanic CO removal (Zafiriou et al., 2003).

The global oceanic source was assessed based on successive oceanic cruises. Linnenbom et al. (1973) first estimated a

flux to the atmosphere of 94 Tg C yr$^{-1}$ by extrapolating data from the Arctic and the North Atlantic and Pacific Ocean. Later extrapolations from *in situ* measurements lead to global estimates ranging between 3 and 600 Tg C yr$^{-1}$ (Logan et al., 1981; Conrad et al., 1982; Bates et al., 1995; Springer-Young et al., 1996; Zuo and Jones, 1995). Such a range reflects the scarcity of the available data given the spatial and temporal heterogeneity of processes controlling oceanic CO. More recent estimations, using larger data sets from the remote ocean, stand on the lower range of previous estimates: Stubbins et al.

(2006a) proposed a global oceanic flux of 3.7 ± 2.6 Tg C yr$^{-1}$ with Atlantic data and Zafiriou et al. (2003) a flux of 6 Tg C yr$^{-1}$ with the large Pacific data set of Bates et al. (1995). Therefore, the oceanic source of CO seems to play a minor role in the global atmospheric budget of carbon monoxide since global emissions exceed 2000 Tg C yr$^{-1}$ (Duncan et al., 2007; Holloway et al., 2000) and are dominated by combustion processes (fossil fuel use and biomass burning) and secondary chemical production in particular due to $CH_4$ oxidation (Liss and Johnson, 2014). Nevertheless, oceanic emissions of CO may play a





role in the remote ocean as it can regionally impact the oxidizing capacity of the troposphere. The only geographical distribution of CO for use in global atmospheric chemistry models was derived from a single model estimation (Erickson, 1989) based on a simple relationship relating the incoming radiation to CO concentration in surface waters. It constitutes to date the only spatialized data easily accessible for the atmospheric modeling community (see ECCAD database, 2018).

Hence the oceanic natural source needs to be better characterized in terms of process and amplitude.

The present work proposes to assess the marine source of CO using a global 3-D oceanic biogeochemical model, in combination with an original dataset gathering *in situ* measurements of oceanic CO performed over the last 50 years. The main CO production and removal processes are explicitly added to the NEMO-PISCES (*Nucleus for European Modelling of the Ocean*, Madec et al. 2008, *Pelagic Interaction Scheme for Carbon and Ecosystem Studies*, Aumont et al., 2015) model. It

allows to characterize the seasonal and spatial variability of CO at the global scale. Then, different experiments, exploring the parameterizations of the main processes, are presented. The resulting simulated oceanic CO concentrations are compared to the dataset of observed concentrations. Finally, an updated spatio-temporal distribution of oceanic CO emissions is proposed for use in current tropospheric chemical models.

## 2 Methods

### 2.1 Oceanic CO model description

The oceanic CO sources and sinks are computed using the global ocean biogeochemistry model PISCES (*Pelagic Interaction Scheme for Carbon and Ecosystem Studies,* Aumont et al., 2015). The PISCES version used in this study (version 2) is described and evaluated in details in Aumont et al. (2015) and we only recall here its main characteristics. PISCES includes 24 tracer variables, with two phytoplankton types (nanophytoplankton and diatoms), two zooplankton size-classes

(micro and mesozooplankton), two organic particles size-classes and semi-labile dissolved organic matter. It also includes five nutrients ($NO_3$, $NH_4$, $PO_4$, Si and Fe), as well as a representation of the inorganic carbon cycle. Phytoplankton growth is limited by light, temperature and by the 5 limiting nutrients. The evolution of phytoplankton biomass is also influenced by mortality, aggregation and by grazing. Chlorophyll (Chla) concentrations for the two phytoplankton types are prognostically computed using the photo-adaptative model of Geider et al. (1996), with Chla/C ratio varying as a function of light and

nutrient limitation.

A specific module has been added to the PISCES – version 2 model in order to explicitly represent the currently identified oceanic CO sources and sinks. These sources and sinks are the photoproduction ($Prod_{Photo}$), the phytoplanktonic production ($Prod_{Phyto}$), the bacterial consumption ($Cons_{Bact}$) and the flux to the atmosphere ($Flux_{ocean\text{-}atmo}$). They affect the oceanic CO concentration according to:

$$\frac{dCO}{dt} = Prod_{Photo} + Prod_{Phyto} - Cons_{Bact} - Flux_{ocean-atmo} \qquad (1)$$





### 2.1.1 Photoproduction

The photoproduction rate is driven by light, the quantity of organic matter bearing chromophoric function (CDOM) and the probability for excited CDOM to produce CO. It is a strongly wavelength-dependent process and according to Fichot and Miller (2010), the relevant range for CO photoproduction is 290-490 nm with a maximum production around a
wavelength (referred as λ in nm) of 325 nm. Hereafter, we describe our main hypotheses to compute the photoproduction.

*Spectral solar irradiance*

We first derive the spectral solar irradiance in the range 290-490 nm ($E_{co}(\lambda,0)$ in W m$^{-2}$) reaching the surface of the ocean as a fraction $f_{co}(\lambda)$ of the total solar irradiance reaching the considered gridbox ($E_{tot}$):

$$E_{CO}(\lambda, 0) = f_{CO}(\lambda) \times E_{tot} \tag{2}$$

For each wavelength this fraction has been determined using the standard solar spectrum ASTM G173-03 (2012). The spectra are modeled using the ground-based solar spectral irradiance SMARTS2 (version 2.9.2) *Simple Model for Atmospheric Transmission of Sunshine*. Only the incident irradiance is considered to determine the quantity of photons affecting CDOM since the upwelling irradiance has been shown to be negligible even in the presence of reflective sediments (Kirk, 1994). This assumption widely simplifies the computation of photochemistry in the ocean. Irradiance then decreases
with depth $z$ (in m), with seawater attenuation coefficients $k$ (in m$^{-1}$) depending on both λ and on Chla concentration according to the relation:

$$E_{CO}(\lambda, z) = E_{CO}(\lambda, 0) \times exp \left[ - \int_z (k(\lambda, Chla)dz) \right] \tag{3}$$

The attenuation coefficients are computed from the coefficients for pure water $K_w$ and biogenic compounds $K_{bio}$ according to Morel and Maritorena (2001):

$$k(\lambda, Chla) = K_w(\lambda) + K_{bio}(\lambda, Chla) \tag{4}$$

$$K_{bio}(\lambda, Chla) = \chi(\lambda)[Chla]e^{(\lambda)} \tag{5}$$

where the coefficients $K_w$, $\chi$ and $e$ as published by Morel and Maritorena (2001) are known for wavelengths ranging between
350 nm and 800nm. A linear extrapolation of the coefficients is performed to retrieve coefficients between 290 and 350 nm.

*CDOM content*

The CDOM content is usually characterized by the absorption coefficient ($a_{cdom}(\lambda,z)$ in m$^{-1}$) at a given λ. For case 1 waters, i.e. far from terrestrial runoff and terrigenous influence, the CDOM is essentially composed of products released
during the initial photosynthetic process (Morel, 2009), and hence $a_{cdom}$ and Chla co-vary (Morel and Gentili, 2009). We use here the Morel (2009) parameterization, which relates $a_{cdom}$(*400 nm*) to Chla (in mg m$^{-3}$):



$$a_{cdom}(400) = 0.065[Chla]^{0.63} \tag{6}$$

The $a_{cdom}(\lambda,z)$ values for each wavelength between 290 and 490 nm are then exponentially extrapolated from $a_{cdom}(ref=400nm)$, with S = - 0.018 nm$^{-1}$ (Morel, 2009) (Fig. 1):

$$a_{cdom}(\lambda) = a_{cdom}(ref) \times e^{(-S(ref-\lambda))} \tag{7}$$

### *Efficiency of the excited CDOM to produce CO*

When the CDOM absorbs photons, only a small and variable fraction of the excited CDOM leads to a photochemical reaction. This fraction is called the apparent quantum yield (hereafter AQY in moles of CO produced / moles of photons absorbed by CDOM). We assume that CDOM is homogeneous in term of composition and thus consider a unique spectral distribution of AQY. We compute the spectral variation of AQY by taking the average of two published parameterizations, Ziolkowski and Miller (2007) (Eq. 8) and Zafiriou et al. (2003) (Eq. 9a and 9b) (Fig. 1):

$$AQY(\lambda) = \exp[-9{,}134 + 0{,}0425(\lambda - 290)] + \exp[-11{,}316 + 0{,}0142(\lambda - 290)] \tag{8}$$

$$AQY(\lambda) = (5{,}78 \times 10^{-6}) \times \exp[-0{,}05(\lambda - 360)] - (6{,}99 \times 10^{-7}) \quad \lambda < 360\ nm \tag{9a}$$

$$AQY(\lambda) = (5{,}24 \times 10^{-6}) \times \exp[-0{,}0229(\lambda - 360)] \quad \lambda > 360\ nm \tag{9b}$$

These empirical parameterizations were derived statistically from measurements of AQY spectra made on seawater samples collected in the Gulf of Maine, the Sargasso Sea, and the North-West Atlantic waters for Ziolkowski and Miller (2007), and during a transect carried out between 70° S and 45° N in the Pacific Ocean for Zafiriou et al. (2003).

### *Photoproduction*

Finally, the resulting CO photoproduction term is computed by integrating over the spectrum of photochemically active solar radiation, the product of three terms (solar irradiance, CDOM absorption and AQY):

$$Prod_{Photo} = \int_{\lambda=290}^{\lambda=490} E_{CO}(\lambda, z) \times a_{cdom}(\lambda, z) \times AQY(\lambda) \times \frac{\lambda}{hc} \times 10^{-3} d\lambda \tag{10}$$

$\lambda / hc$ converts moles of photons to W×s (Joules), with $h$ the Planck's constant (6.6260755×10$^{-34}$ J×s) and $c$ the speed of light in a vacuum (3.00×10$^8$ m s$^{-1}$).

### 2.1.2 Direct phytoplankton production

The direct production of CO by phytoplankton is not well understood. So far, only one study assesses the direct CO production rates from phytoplankton (Gros et al., 2009), by experimentally exposing different micro-algal species to a diurnal cycle of 12 h of light and 12 h of dark conditions. The measured rates were highly variable from one species to another (from 19 to 374 $\mu$mol CO g Chla$^{-1}$ d$^{-1}$ for diatoms and from 6 to 344 $\mu$mol CO g Chla$^{-1}$ d$^{-1}$ for non-diatoms). Here,





we used the median values of Gros et al. (2009) reported in Tran et al. (2013): 85.5 ($k_{nano}$) and 33.0 µmolCO gChla$^{-1}$ d$^{-1}$ ($k_{diat}$) respectively for the nanophytoplankton and the diatom types. The direct CO production by phytoplankton is thus computed using the relation:

$$Prod_{Phyto} = \frac{h}{12} \times (k_{nano}Chla_{nano} + k_{diat}Chla_{diat}) \tag{11}$$

with $Chla_{nano}$ and $Chla_{diat}$ (in g Chla L$^{-1}$) the respective Chla concentrations for nanophytoplankton and diatoms. Note that we do not explicitly resolve the diurnal cycle in PISCES. Hence, to account for the variation of the day light length, the production rates are divided by 12 hours for conversion in $\mu$mol CO g Chla$^{-1}$ h$^{-1}$ and then multiplied by the number of hours of light $h$, computed as a function of latitude and the period of the year.

### 2.1.3 Bacterial consumption

The bacterial consumption of CO has been studied for the North Atlantic and for polar waters (Xie et al., 2005; 2009). According to these authors, it follows a first-order to zero-order chemical kinetic, or kinetics with a saturation threshold. The values of the parameters associated to these chemical kinetics are highly variable and depend on the environment and water masses characteristics. For a global application in PISCES, we chose a first-order chemical kinetic to reduce the number of uncertain parameters such as:

$$Cons_{Bact} = k_{CO} \times CO \tag{12}$$

with $k_{CO}$ the rate of bacterial consumption (0.2 d$^{-1}$) and CO the concentration of CO (mol L$^{-1}$).

### 2.1.4 Ocean-atmosphere CO exchanges

The CO flux at the ocean-atmosphere interface is described in a similar way to the Fick's diffusion law. It depends on the concentration at the ocean surface and on the partial pressure in the atmosphere above the ocean ($pCO_a$ in atm):

$$Flux_{ocean-atmo} = k_{flx}(CO - H \times pCO_a) \tag{13}$$

$$pCO_a = p_{atm} \times f_{CO} \tag{14}$$

with $p_{atm}$ the atmospheric pressure and $f_{CO}$ the atmospheric mixing ratio. In PISCES, the atmospheric CO concentration over the ocean is considered as spatially constant and fixed to 90 ppbv, closed to its global surface average (Novelli et al., 2003 and ESRL NOAA-GMD website, 2018). $H$ is the Henry's constant, which relates the partial pressure of a gas with the equilibrium concentration in solution ($CO_w*$):

$$CO_w* = H \times pCO_a \tag{15}$$

It is calculated from Weisenburg and Guinasso (1979) by:

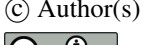



$$\ln H = -169{,}4951 + 263{,}5657 \left[\frac{100}{T}\right] + 159{,}2552 \ln\left[\frac{T}{100}\right] - 25{,}4967 \left[\frac{T}{100}\right] + S‰ \left[0{,}051198 - \right.$$
$$\left. 0{,}044591 \left[\frac{T}{100}\right] + 0{,}0086462 \left[\frac{T}{100}\right]^2\right] \tag{16}$$

with S‰ the salinity in parts per thousand. Finally, $k_{flx}$ is the gas transfer velocity (in m s$^{-1}$), which depends on the Schmidt number $sch$ (Zafiriou et al., 2008), the temperature (T in °C) and the wind speed at 10 m high ($u$ in m s$^{-1}$) (Wanninkhoff, 1992):

$$k_{flx} = [0{,}251 \times u^2] \times \left[\frac{660}{sch}\right]^{1/2} \tag{17}$$

$$sch = -0{,}0553T^3 + 4{,}3825T^2 - 140{,}07T + 2124 \tag{18}$$

5 ## 2.2 Tests of alternative parameterizations

Alternative parameterizations have been tested on the CDOM absorption coefficient and on the bacterial consumption rate. First because other parameterizations for these terms exist in the literature, and second because the photoproduction and the bacterial sink are the major processes controlling the oceanic CO. The table 1 summarizes these experiments.

10 *CDOM parameterization*

Regarding the CDOM absorption coefficient as a function of Chla, two parameterizations were tested in addition to the one of Morel (2009). The first one was initially developed for the photoproduction of carbonyl sulfide by Launois et al. (2015). It relates $a_{cdom}$ at 350 nm to the log of Chla ($C$ in mg m$^{-3}$):

$$\ln(a_{cdom}(350)) = 0{,}5346C - 0{,}0263C^2 - 0{,}0036C^3 + 0{,}0012C^4 - 1{,}6340 \tag{19}$$

This relation has been derived from monthly climatologies of Chla concentrations and surface reflectances obtained with the 15 MODIS Aqua ocean color between July 2002 and July 2010 (Maritorena et al., 2010; Fanton d'Andon et al., 2009). Using these reflectances, the SeaUV algorithm developed by Fichot et al. (2008) allows to calculate the attenuation coefficients $K_d$ at 320 nm. Then a ratio $a_{cdom}(320) / K_d(320) = 0.68$ obtained from *in situ* measurements permits to calculate $a_{cdom}(320nm)$ (Fichot and Miller, 2010). Finally, Eq. (7) has been used to estimate $a_{cdom}$ at 350 nm. The second other parameterization tested is based on the calculation of $a_{cdom}$ at 440 nm with the relation Garver and Siegel (1998). It has been obtained from 20 computed organic matter absorption and observed Chla concentration:

$$\text{per}(a_{cdom}(440)) = -26[\log(Chla)])] + 26 \tag{20}$$

*per($a_{cdom}(440)$)* is the contribution of CDOM to the total absorbed radiation of colored organic compounds in sea water. It is calculated by:



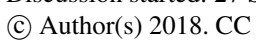

$$per(a_{cdom}(440)) = \frac{a_{cdom}(440)}{a_{ph}(440) + a_{cdom}(440)} \times 100 \qquad (21)$$

with $a_{ph}(440)$ the absorption of phytoplankton, proportional to the Chla (Preiswerk and Najjar, 2000):

$$a_{ph}(440) = 0,0448 \times Chla \qquad (22)$$

Equation (7) is then used to calculate $a_{cdom}$ at each wavelength. The variations of the CDOM absorption with Chla for all the parameterizations we tested, including the relation of Morel (2009), are shown in Fig. 2.

*Bacterial consumption rate*

The rate of CO bacterial consumption is highly uncertain. According to experimental measurements, it seems to vary from several orders of magnitude. For example, Zafiriou et al. (2003) measured consumption rates of the order of 0.1 $d^{-1}$ in the Southern Ocean, while Day and Faloona (2009) measured rates of more than 19 $d^{-1}$. Hence, tests have been carried out on this rate. It is first considered constant with values ranging from 0.1 to 1.0 $d^{-1}$. Secondly, it is considered to vary with the

intensity of phytoplanktonic activities (proportional to Chla) and the

water temperature, according to the study of Xie et al. (2005):

$$k_{CO} = 24 \times \mu \times [A(T + 2)] \times chl + Y \qquad (23)$$

with $chl$ the Chla concentration in $\mu$g $L^{-1}$ et $T$ the temperature in °C. $A$ and $Y$ are constant parameters (0.0029 and 0.16 respectively). This equation was derived from dark incubations of 44 water samples from the Delaware bay, Beaufort Sea and North west Atlantic waters in summer and autumn. The study showed a positive correlation between $k_{CO}$, Chla and

temperature ($R^2$=0.69 with $\mu$=1). For its use in the global ocean and to reduce the mean resulting rate to 0.2 $d^{-1}$, we have reduced the calculated rate $k_{CO}$ from the original equation of Xie et al. (2005) by multiplying by a constant $\mu$, fitted to 0.05.

**2.3 Standard experiment**

The modified PISCES version, in which we include the CO module, is coupled to the ocean general circulation model Nucleus for European Modelling of the Ocean version 3.6 (NEMO v3.6). We use the global configuration ORCA2,

with a nominal horizontal resolution of 2° and 30 levels on the vertical (with 10m vertical resolution in the first 200 meters). The initial biogeochemical conditions for all tracers except CO are obtained after a 3000-year long spin-up using NEMO-PISCES as detailed in Aumont et al. (2015). CO concentrations are initialized at zero at the start of year 1 and NEMO-PISCES is then run for two years using the same forcing fields than the ones described in Aumont et al. (2015). The first year is considered as a short spin-up because of the very short life-time of CO in the ocean, and the second year is used for

the analysis presented below. The same procedure is repeated for all alternative parameterizations of the CO module, i.e. a 2-year simulation with the last year used for the analysis.




### 2.4 Comparison to in situ data

We compiled the existing measurements of oceanic CO concentrations in order to evaluate the realism of the model results. Two types of measurements were included in this compilation: measurements of CO in the surface waters and profiles describing the concentrations as a function of depth. This dataset covers fairly well the Atlantic and Pacific Ocean although some wide areas are not documented (such as in the Indian ocean).

This compilation has been designed to be comparable as far as possible to the global model outputs. Indeed, as the model has a rather coarse horizontal resolution and as it does not resolve diurnal variation, we increased the spatial and temporal coverage of the data when necessary. To do so, we took the data described in the literature (which were sometimes already aggregated), and we averaged some of it. However, it is worth mentioning that averaging was not always possible and therefore after treatment each data point potentially includes a different number of observation. Tables 2 and 3 present the different datasets collected for the evaluation and the treatment we performed, respectively for the surface data (14 datasets) and for the profiles (10 datasets). After treatment, 285 surface data points and 24 vertical profiles were analyzed and compared to the model output.

For the surface data, the root mean square error (RMSE in nmol CO L$^{-1}$) has also been calculated:

$$RMSE = \sqrt{\frac{\sum_N (CO_{MOD} - CO_{OBS})^2}{N}} \qquad (24)$$

with N the number of observed concentrations (N = 285 points).

## 3 Results and discussion

### 3.1 The oceanic CO budget

In this section we describe the oceanic CO cycle as simulated by PISCES using CO module implemented in this study. The global oceanic fluxes are first exposed. Then, the spatial patterns of the CO concentration as well as the different sources and sinks terms are presented.

### 3.1.1 Global oceanic fluxes

At the scale of the global ocean, the CO inventory is 0.3 Tg C, with a residence time of 4.3 days. The global photoproduction term in the ocean is of 19.2 Tg C yr$^{-1}$, which is more than 3 times the direct phytoplankton production (6.2 Tg C yr$^{-1}$), and of the same order of magnitude than the bacterial consumption term (21.9 Tg C yr$^{-1}$). The global CO emission to the atmosphere (3.6 Tg C yr$^{-1}$) is rather small compared to the biochemical sources and sinks processes. Figure 3 summarizes the values of the different terms of the CO budget in the oceanic surface layer. Only 0.1 Tg C of CO is contained in the surface layer (considered as the first 10 m of the ocean). Almost one half of the CO photoproduction takes place in the surface layer of the ocean (9.3 Tg C yr$^{-1}$ in the first 10 meters), whereas only 1/20 of the phytoplankton





production unfolds there (0.3 Tg C yr$^{-1}$). Most of the CO produced at the surface is either emitted to the atmosphere or transported vertically towards deeper layers. The mean downward flux from the surface layer (at 10 meters' depth) is estimated to be 2.5 Tg C yr$^{-1}$. It is therefore of the same order of magnitude as the estimated emissions to the atmosphere, which points out the importance of taking into account the effects of ocean circulation and mixing to estimate and map CO

emissions.

### 3.1.2 Spatial patterns of the sources and sinks

Figure 4 presents the annual mean of oceanic CO concentrations and the different sources and sink terms, all vertically integrated on the first 1000 meters. The integrated CO concentrations are highest at low latitudes and tend to

decrease poleward. A strong CO maximum of more than 100 $\mu$mol m$^{-2}$ is reached along the equator in the middle of the Pacific Ocean. Being the main CO source, photoproduction shows a pattern similar to CO concentrations, with a co-localization of their maximum (exceeding 8 mmol m$^{-2}$yr$^{-1}$ for photoproduction). On the contrary, the integrated direct biological production is stronger at mid latitudes, especially in the Southern Ocean where it reaches more than 3 mmol m$^{-2}$yr$^{-1}$ and is weaker in the oligotrophic gyres (around 1 mmol m$^{-2}$yr$^{-1}$). The bacterial consumption term (from 0 to 8 mmol m$^{-2}$yr$^{-1}$

$^{-1}$) is linearly dependent on the CO concentration and is therefore following its spatial pattern.

In the surface layer (taken here as the first 10 meters), the patterns of the annual mean CO concentrations and the related fluxes are slightly different from the vertically integrated ones. Figure 5 presents the surface pattern of CO concentrations and photoproduction. As for the integrated pattern, CO concentration is higher at the equator, with a strong maximum (more than 3 nmol L$^{-1}$) found in the eastern equatorial Pacific Ocean. This CO maximum occurs in a highly

productive area where surface waters are enriched in nutrients by the equatorial upwelling, which stretches from Galapagos Island to the South Equatorial Current and decreases westward (Wyrtki, 1981). CO is there produced by an active photoproduction (up to 1.4 nmol L$^{-1}$ d$^{-1}$) due to both high irradiance (more than 6 W m$^{-2}$) and high level of CDOM, and is also easily concentrated in surface waters with the strong upward currents. Compared to the integrated patterns, the annual mean surface CO concentrations as well as the surface photoproduction present clear minima in the center of the subtropical

gyres. Photoproduction is there around 0.5 nmol L$^{-1}$ d$^{-1}$ and CO concentration less than 1 nmol L$^{-1}$. In the oligotrophic gyres, Chla and hence CDOM are low, which prevents high CO photoproduction. However, light penetrates deeper allowing photoproduction to occur, even at low rate, all along the irradiated water column. In an opposite way, photoproduction in highly productive areas occurs mainly in surface waters as organic materials absorb most of the available irradiance.

### 3.2 Evaluation of the oceanic CO concentration

Simulated surface CO concentrations have been compared to *in situ* concentrations, by extracting the model results collocated in time and space with measurements. Note that the model simulation is here climatological so that the year of measurement cannot be taken into account for the comparisons.





### 3.2.1 Surface CO concentrations

Observed daily mean surface CO concentrations range from 0 to 13.9 nmol $L^{-1}$ with a mean value of 1.9 nmol $L^{-1}$. 75% of the observed data are below 2.5 nmol $L^{-1}$. The simulated concentrations range from 0.3 to 4.2 nmol $L^{-1}$, with a mean of 1.5 nmol $L^{-1}$ for the selected months and locations. The RMSE value resulting from the comparisons is 1.84 nmol $L^{-1}$.

Between 50°N and 50°S, the model is able to represent most concentrations within a factor of two (Fig. 6, C). For instance, the Atlantic Meridional Transect data (Stubbins et al., 2006b) from 45°N to 30°S, present rather constant surface CO concentrations around 1.5 nmol $L^{-1}$ which are particularly well reproduced by the model. In the Pacific Ocean, the data set of Bates et al. (1995) covers a broader range of latitudes and longitudes (Fig. 6, A) and the model is consistent with the measured concentrations. A longitudinal gradient is observed with concentrations significantly higher in the East and Central

Equatorial Pacific as compared to the western part. Bates et al. (1995) attribute this gradient to increased biological productivity, Chla and CDOM in the eastern upwelling region. Second, Bates et al. (1995) data show a clear latitudinal pattern with higher concentrations from either side of the 15-30°S oligotrophic band (Fig. 6, B). This pattern has also been highlighted by Swinnerton and Lamontagne (1973) who sampled from 20°N to more than 60°S. However, their high values at the Equator (up to 6.3 nmol $L^{-1}$) are much higher than the ones of Bates et al. (3.0 nmol $L^{-1}$). Finally, the data set of Bates

et al., which covers different periods of the year, points out the seasonal variability of the concentrations at mid-latitudes, in agreement with the photochemical nature of the main CO source.

At latitudes higher than 50° north or south, the model underestimates the reported high daily mean surface CO concentrations (Fig. 6, B). In the Southern Ocean, simulated surface concentrations do not exceed 4.5 nmol $L^{-1}$, whereas the data of Swinnerton and Lamontagne (1973), sampled in December, can reach 8.9 nmol $L^{-1}$. The Southern Ocean is also the

basin where Bates et al. (1995) reported their highest surface values (up to 4.7 nmol $L^{-1}$). In addition, their concentration is higher for December (3.5 nmol $L^{-1}$) than for March and April (0.8 nmol $L^{-1}$). Therefore, the underestimation seems to occur mainly during the phytoplankton bloom season, indicating a possible bias in the CO production processes. In the Arctic, we compared simulated concentrations with the datasets of Xie et al. (2009) and Tran et al. (2013). Xie et al. (2009) measured in the open waters of the southeastern Beaufort Sea, a mean CO concentration of only 0.4 nmol $L^{-1}$ in autumn which is

particularly well represented by the model. However, their higher mean value of 4.7 nmol $L^{-1}$ during the spring season is highly underestimated, indicating once again a possible bias with the CO production processes. Tran et al. sampled in July 2010 between Svalbard, Greenland and Iceland. Closer from the Norwegian basin, measurements have been performed in open ocean water. It presents there a high variability (from 1.4 to 8.7 nmol $L^{-1}$) but most data points are under 4.0 nmol $L^{-1}$ and represented within a factor of 2 by the model. Closer from the Greenland shelf, measurements have been performed in

polar waters where pack ice was present. In this area, measured surface concentrations are significantly higher (up to 13.9 nmol $L^{-1}$) and are clearly underestimated by the model. Other studies have measured high surface CO concentrations in polar regions (Xie and Gosselin, 2005; Gros et al., personal communication), which tends to strengthen the conclusion that major mismatches between our modeled and the observed CO concentrations occur in that region. Indeed, Gros et al. (personal communication) measured, north of Svalbard, a mean concentration of 7.1 nmol $L^{-1}$ in May 2015. These high CO





concentrations reported in polar region could be due to the release of organic matter in the open ocean during ice melting due to algae growing in the ice (Belzile et al., 2000). It could also be due to the direct release of CO produced inside the ice. Indeed, Xie and Gosselin (2005) mentioned that sea-ice is a suitable matrix for efficient photo-reactions involving CDOM, and measured concentrations in May exceeding 100 nmol L$^{-1}$ in the bottom layer of a few first-year sea-ice cores sampled in

the southeastern Beaufort Sea. However, according to Xie et al. (2009) and Zafiriou et al. (2003), the high CO concentrations reported at high latitudes may be mainly due to slower microbial CO uptake rates in cold waters. Indeed, Zafiriou et al. estimated rates as low as 0.09 d$^{-1}$ in the Southern Ocean.

It is the first time that such a dataset of *in situ* CO concentrations in the surface ocean have been gathered and that a 3-D global biogeochemical model is used to simulate the oceanic CO cycle. Despite the scarcity of measurements regarding in

space and time, PISCES reproduces the observed surface concentrations reasonably well, at least for the open ocean between 50°N and 50°S. While concentrations at low and mid latitudes are simulated within a factor of 2, photoproduction or consumption processes might not be well represented at high latitude.

### 3.2.2 Vertical CO profiles

In this section, we evaluate the vertical distribution of CO concentrations as simulated by PISCES against available observed vertical profiles. Only 10 available profiles upon 24 are shown hereafter (Fig. 7), covering different types of marine environments. Other ones are available in Supplementary section.

All simulated CO profiles exhibit a decrease with depth. They are mainly driven by the photoproduction source which is a combination of the CDOM and irradiance levels. Indeed, no clear sub-surface maximums, potentially associated to direct

phytoplankton production, is seen. This is also the case for the observed profiles. At 100 meters, simulated as observed concentrations reach values close to zero, explained by the negligible influence of light irradiance under these depths. However, the shapes of the decreases differ with time and locations, which can be related both to differences in the light penetration and mixed layer depths.

When trying to compare simulated and observed profiles one by one, we observe quantitative differences. When the

model is able to correctly simulate the concentration at the surface, the deeper concentration is also well simulated. On the contrary, when the model over or under estimate the surface concentration, it is also the case below (see as an example the representation of the two profiles of Zafiriou et al., 2008 taken in the Sargasso Sea in March and April). Nevertheless, the model is always in good agreement with the nearly zero concentrations below 100 meters. It is however particularly hard to bring out spatial or temporal trends for the observed under or over estimations. As an example for the equatorial Pacific

region, the model underestimates with a factor of 2 the concentrations of Johnson and Bates (1996) whereas in the same region in November the data of Otha (1997) are well represented. Additionally, it is worth mentioning that high variability and uncertainty is associated to some measurements. For example, between 25 and 50 meters Yang et al. (2011) measured for spring month concentrations from 0.0 to 7.5 nmol L$^{-1}$ in the Coastal East China and Yellow Seas. Also, when no standard deviation is available the data reflects one-time measures. Especially, this is the case for the 10 profiles of Swinnerton and




Lamontagne (1973), taken during early afternoon. It is hence difficult to compare these profiles with monthly averaged model outputs.

### 3.3 Sensitivity to alternative parameterizations

5       Alternative parameterizations have been tested on the photoproduction and the bacterial consumption term, which are respectively the main source and sink for oceanic CO.

### 3.3.1 CO photoproduction

*Surface CDOM absorption*. Different parameterizations for the CDOM absorption coefficient have been tested, all using the same simulated Chla concentrations. The latitudinal mean of these coefficients, taken at 443 nm and for the surface ocean, are shown on Fig. 8. The mean climatology of CDM (Colored Detrital Matter) distributed within the GlobColour products (Maritorena et al., 2010; Fanton d'Andon et al., 2009), averaged over 2002-2012, is shown as well. They all lead to similar distributions with higher values at high latitudes, intermediate ones near the Equator and the lowest in the subtropical gyres,

as it is the case for Chla surface concentrations. However, the magnitudes of these CDOM absorption coefficients are very different and vary within a factor of 3 across the different parameterizations. The one of Morel (2009), chosen for our standard simulation, gives intermediate values (global mean is 0.014 m$^{-1}$) between Preiswerk and Najjar (mean 0.008 m$^{-1}$) and Launois et al. (mean 0.026 m$^{-1}$). The GlobColour CDM content can be considered as a proxy for CDOM as it has been postulated that the CDOM absorbance represents 90% of the CDM one (Siegel et al., 2002). Latitudinal variations are

consistent between the satellite products and the tested parameterizations. Minimums (less than 0.01 m$^{-1}$) are located in oligotrophic gyres. Quantitatively, those minimums are best represented by the relation of Morel, since the ones of Launois et al. (2015) and of Preiswerk and Najjar (2000) give too high and too low values, respectively. Highest satellites-derived CDM absorptions are reached at high latitudes, especially in the northern hemisphere (latitudinal mean goes up to 0.060 m$^{-1}$ between 50 and 80°N). These high values correspond mainly to coastal areas, and none of the parameterizations are able to

reproduce them. It can be explained first by the fact that our simulation does not properly resolve the coastal zones due to its too coarse resolution (Aumont et al., 2015). Second, the parameterizations are all depending on Chla and are therefore probably better suited for Case-1 waters (Matsushita et al., 2012), for which most dissolved organic matter comes from local phytoplanktonic activities like cell lyses, exsudation or grazing (Para et al., 2010). Indeed, the CDOM concentration in coastal waters is also regulated by the interactions with the continent (Bricaud et al., 1981; Siegel et al., 2002), especially at

rivers mouths where CDOM concentrations are generally higher than for the rest of the ocean (Para et al., 2010). This is the consequence of a local stimulation of primary production associated to nutrient supply (Carder et al., 1989), but also of a direct supply of continental CDOM (Nelson et al., 2007). This direct CDOM supply is not represented in the model, potentially leading to an underestimation of the CO photoproduction in coastal areas, and potentially also in open ocean



waters under terrestrial influence. This is particularly the case for the Arctic ocean as it is enriched in terrestrial dissolved organic matter due to high river supply (Dittmar and Kattner, 2003). Additionally, this could explain part of the CO underestimations in the Arctic, together with the aforesaid mechanism associated to the presence of sea ice. In the Austral Ocean, however, whereas CO underestimations were observed, the CDOM content is slightly above the CDM content

retrieved form space (except at the very close Antarctic coast). Hence, we better opt in this region for a link with the representation of the bacterial consumption term. Finally, among the three parameterizations, we chose the one of Morel (2009) for our representation of the CDOM absorption despite the underestimations in the northern hemisphere, because it globally better fits the satellite-derived CDM as the amplitude of the latitudinal gradient is the largest (standard deviation is 0.007 m$^{-1}$, against 0.005 m$^{-1}$ for Launois and 0.003 m$^{-1}$ for Preiswerk and Najjar relations). Nevertheless, it is important to

mention that CDOM is a very heterogeneous matter and therefore using Chla as a proxy for CDOM is in any case obviously reductive, as CDOM has its own dynamic.

***Sensitivity of the oceanic CO budget to changes in the photoproduction.*** Global fluxes have been calculated from the simulations using the different CDOM parameterization (table 1, lines 1-3). The global photoproduction flux is more than

doubled when using the parameterization of Launois et al. (49.1 Tg C yr$^{-1}$ against 19.2 Tg C yr$^{-1}$ when using Morel) and decreases with the one of Preiswerk and Najjar (14.2 Tg C yr$^{-1}$), in agreement with the quantitative tendencies of the CDOM surface values discussed above. For these different tests, the bacterial consumption rates have been kept the same so that the changes in the CO budget and in other terms are solely due to changes in photoproduction. The CO flux to the atmosphere (2.7-9.0 Tg C yr$^{-1}$) and the bacterial sink (17.9-46.5 Tg C yr$^{-1}$) both vary accordingly with the change in photoproduction. As

a direct consequence, the oceanic CO total inventory is modified (0.2-0.6 Tg C), which also changes the values of the RMSE when comparing simulated surface CO concentrations with *in situ* measurements (it is 2.64 nmol L$^{-1}$ with Launois et al. and 1.98 nmol L$^{-1}$ with Preiswerk and Najjar whereas it is 1.84 nmol L$^{-1}$ with Morel). Note that the direct phytoplankton production term remains the same as it does not depend on CO concentrations. Photoproduction has also been integrated in the mixed layer in order to be compared to the value of 39.3 Tg C yr$^{-1}$ computed by Fichot and Miller (2010) with a global

model using ocean color data for the CDOM parameterization. With a budget of 45.4 Tg C yr$^{-1}$, the photoproduction of Launois et al. gives the closest estimation, the ones resulting form Morel and from Preiswerk and Najjar being much lower (17.7 Tg C yr$^{-1}$ and 13.1 Tg C yr$^{-1}$ respectively). Those two estimations are also lower than the most recent estimations based on extrapolations of *in situ* measurements. Indeed, Zafiriou et al. (2003) estimated the photoproduction between 30 and 70 Tg C yr$^{-1}$ based on a large data set collected in the Pacific Ocean and Stubbins et al. (2006b) estimated it between 38 and 84

Tg C yr$^{-1}$ with Atlantic Ocean data.

### 3.3.2 CO consumption

Simulations regarding the bacterial consumption have been performed. First, different values for a constant bacterial rate have been tested. Second, a variable rate as a function of Chla and temperature according to the relation of Xie et al.



2005 has been tested. Global budgets resulted from these different tests are shown in table 1 (lines 4-8), using the same CDOM parameterization (Morel 2009) so that the impact of the photoproduction term on the oceanic CO concentration remains the same. When increasing the rate from 0.1 to 2.0 d$^{-1}$ (with a constant value), the bacterial sink increases from 19.4 to 25.4 Tg C yr$^{-1}$. This rise of consumption induces a strong decrease of the CO inventory (from 0.5 to 0.0 Tg C) and

therefore also decreases the flux to the atmosphere (from 5.6 to 0.4 Tg C yr$^{-1}$). When changing the rate, the fit of the surface CO concentration with observations is also modified. Indeed, when it is 0.1 d$^{-1}$, the RMSE value is increased (=1.93 nmol L$^{-1}$) compared to the control run (RMSE=1.84 nmol L$^{-1}$) because of an overall overestimation of the concentrations. When the consumption rate exceeds 0.2 d$^{-1}$, the RMSE value is also increased but that time because of an overall underestimation. For the test using a spatio-temporal varying rate (table 1, line 8), we slightly modified the initial equation of Xie et al. (2005) by

adding a factor $\mu$=0.05, so that the mean $k_{CO}$ is 0.2 d$^{-1}$. Indeed, when using the original $\mu$=1 with our model, bacterial rates vary from 3.8 to 7.0 d$^{-1}$ and thus lead to a strong underestimation of the CO concentrations. When a varying rate is applied, the global bacterial sink, the CO inventory and the CO flux to the atmosphere are pretty much the same than using the standard run with a constant rate of 0.2 d$^{-1}$. Also, the RMSE value is slightly the same (1.81 nmol L$^{-1}$). Thus, applying this varying rate did not help to improve the fit of CO concentrations against in situ measurements, which could be explained by

the very low standard deviation of the resulting $k_{CO}$ values (0.01 d$^{-1}$).

Being the main CO sink in the ocean, a more accurate parameterization of the bacterial CO consumption term is essential to quantify CO emissions to the atmosphere. However, little is known on the CO bacterial consumption, although it has been shown to be ubiquitous in diverse marine ecosystems, particularly in the North East Atlantic and Arctic waters (Xie et al., 2005; 2009). According to Xie et al. (2005), this process could follow different patterns from first to zero-order

kinetics or saturation, but most marine waters may be reasonably well approximated by the first-order kinetic as CO concentrations rarely exceed the half saturation constant value. It is nevertheless difficult to constrain the bacterial consumption rate based on experimental estimations as they present a very high variability. Highest rates seem to be found for coastal waters, suggesting the presence of active CO oxidizing communities (Tolli et al., 2006; Tolli and Taylor, 2005). For example, Day and Faloona (2009) measured rates from 1.2 to 19.2 d$^{-1}$ along the Californian coasts and Jones and

Amador (1993) rates from 0.2 to 12 d$^{-1}$ in the Caribbean Sea. In the remote ocean, measured rates are lower but the variability remains high. Zafiriou et al. (2003) measured a rate as low as 0.1 d$^{-1}$ in the Southern Ocean, whereas Conrad et al. (1982) retrieved values around 0.7 d$^{-1}$ in the Equatorial Atlantic and Otha (1997) more than 3.0 d$^{-1}$ in the Equatorial Pacific. Part of this high variability might be explained by the fact that as for other microbial processes, the CO consumption should also depend on a number of parameters like microbial species, productivity, temperature or ocean acidity (Xie et al. 2005).

Using the only study proposing to compute the $k_{CO}$ value with Chla and temperature (Xie et al. 2005), we attempted to account for this variability. However, it seems that such an empirical equation, developed with data from the Delaware Bay, the North West Atlantic and the Beaufort Sea, is not suitable for a global application as it leads to very high $k_{CO}$ values. Even when tuning with the factor $\mu$, the remaining $k_{CO}$ variability is too small to significantly improve the fit with the observed surface CO concentrations.



### 3.4 Simulated CO emissions

This section first presents the CO emissions to the atmosphere resulting from our standard simulation. Then, the emissions are compared to the canonical distribution of CO emissions published by Erickson (1989), which is the only gridded global CO emission dataset available to date in the literature.

Figure 9 presents the spatial patterns of emissions resulting from PISCES (panel A). All oceanic regions are net sources of CO for the atmosphere, with emissions varying spatially from 0 to 2.1 mmol $m^{-2}$ $yr^{-1}$ on an annual mean basis with a global mean flux of 0.7 mmol $m^{-2}$ $yr^{-1}$. The spatial pattern of emissions follows the one of the surface CO concentrations. The strongest emissions are simulated in the equatorial region, especially in the East Pacific Ocean on both sides of the equator. High emissions are also reached locally along the west coast of South America and Africa. Annual mean emissions

are reduced to roughly 0.5 mmol $m^{-2}$ $yr^{-1}$ in the center of the subtropical gyres and to almost zero at latitudes higher than 60°. Simulated CO emissions also present a strong seasonal variability (Fig. 9, panel B). Whereas emissions at the equator (30°S-30°N) are roughly constant throughout the year (around 1.0 mmol $m^{-2}$ $yr^{-1}$), emissions at intermediate latitudes (30° to 60°) vary seasonally from 0 to the highest values encountered (up to 2.0 mmol $m^{-2}$ $yr^{-1}$). However, on a yearly basis, the equatorial region is the major contributor with a yearly flux of 2.2 Tg C, which is more than 60% of the global value (3.6 Tg C $yr^{-1}$).

Intermediate latitudes contribute to 1.3 Tg C $yr^{-1}$ and polar regions (more than 60°N/S) to only 0.1 Tg C $yr^{-1}$. Additionally, emissions are stronger in the southern hemisphere than in the northern hemisphere (2.1 Tg C $yr^{-1}$ and 1.5 Tg C $yr^{-1}$ emitted respectively), which could be attributable to the larger surface area covered by oceans and to the slightly closer proximity of the Earth to the Sun during the southern summer solstice. Finally, the global emission of 3.6 Tg C $yr^{-1}$ simulated with PISCES fall well into the range of previous estimations based on extrapolations from *in situ* oceanic CO measurements:

Stubbins et al. (2006a) estimated a yearly flux to the atmosphere of 3.7 ± 2.6 Tg C $yr^{-1}$ from Atlantic data, Zafiriou et al. (2003) a flux of 6 Tg C $yr^{-1}$ and Bates et al. (1995) estimated the emissions in the range 3-11 Tg C $yr^{-1}$ with Pacific data.

The CO emissions produced by Erickson (1989) are still currently used as spatial distribution by global chemistry-climate models. They are presented here as published in 1989, but are now generally rescaled to lower total emissions for use in atmospheric models. Figure 9 presents the spatial patterns and seasonal variability of emissions resulting from Erickson's

model (panels C and D respectively). In agreement with our simulation, all oceanic regions are sources of CO for the atmosphere. However, emissions present very large spatial and quantitative differences. On an annual mean basis, Erickson's emissions vary from 0 to more than 33 mmol $m^{-2}$ $yr^{-1}$ (mean is 9.0 mmol $m^{-2}$ $yr^{-1}$) which is one order of magnitude above the ones of PISCES. Emission pattern for Erickson presents clearly two bands of intense outgassing at mid latitudes (between 30° and 60° N/S), whereas pattern for PISCES presents maxima around the equatorial zone and minima in the center of the

oligotrophic gyres. When looking at the seasonal variability, Erickson's model also simulates the highest emissions at mid latitudes in summer months but with much stronger maximum values (up to 40 mmol $m^{-2}$ $yr^{-1}$). Globally, Erickson's model produces a flux of 70.7 Tg C $yr^{-1}$, which is 20 times greater than the PISCES estimation. PISCES flux is therefore much closer from the most recent estimations (Stubbins et al., 2006a; Zafiriou et al., 2003; Bates et al., 1995).



The spatial and quantitative differences between Erickson's and PISCES's emissions are mainly attributable to the fact that the bio-chemical processes known to control the CO concentration at the surface ocean are not accounted for in the Erickson's representation. On a similar way to Eq. (13) of PISCES, the emissions of Erickson depend on the CO concentration gradient between the atmosphere and oceanic part. However, the oceanic part is not computed dynamically and linearly depends on the total available radiation at the surface ocean based on a relation derived empirically with Atlantic data of Conrad et al. (1982). In Fig. 10 is shown the mean annual surface CO concentrations as a function of the mean annual total solar radiation in PISCES, as well as the linear relation used by Erickson. Mean surface radiation in PISCES ranges from 0 to 300 W m$^{-2}$ and the stronger it is, the higher the concentrations for both models. However, for a same given radiation there is in PISCES a multitude of possible values for the CO concentration as it is controlled by different processes (light, CDOM content, Chla, bacterial consumption) and not only by the light intensity. As a result, the mean annual surface concentrations do not exceed 6.6 nmol L$^{-1}$ whereas it can reach 18.0 nmol L$^{-1}$ with 300 W m$^{-2}$ with the Erickson's relation. Values reached by Erickson's model are therefore much higher and this implies stronger outgassing to the atmosphere. It's worth mentioning that given the *in situ* measurements of surface CO concentrations presented in section 3.2, the relation of Erickson would easily overestimate the concentrations for the regions with high radiation.

**4 Conclusion**

We used a global 3-D biogeochemical model to explicitly represent the oceanic CO cycle based on the up-to-date knowledge of its bio-chemical sources and sinks. With our best-guess modelling setup, we estimate a photoproduction of 19.2 Tg C yr$^{-1}$ and a bacterial consumption of 21.9 Tg C yr$^{-1}$. The estimation of the CO flux to the atmosphere is 3.6 Tg C yr$^{-1}$ and falls well into the range of previous recent estimates (Stubbins et al., 2006a; Zafiriou et al., 2003; Bates et al., 1995). The global downward flux at 10 m depth, estimated at 2.5 Tg C yr$^{-1}$, points out the importance of taking into account the effects of ocean circulation and mixing to model the oceanic CO cycle and the interplay between sources / sinks processes and emissions to the atmosphere.

The distribution of surface CO concentrations reflects primarily the distribution of CDOM in the surface ocean and its impact on the photoproduction source, with high concentrations of CO in the biologically productive regions of the ocean. For the first time, a large data set of *in situ* CO measurements collected from the literature has been gathered and used to evaluate these concentrations. Despite the scarcity of these measurements in terms of their spatial and temporal distribution, the proposed best simulation is able to represent most of the ~300 data points within a factor of two.

That said, there are a number of limitations that probably preclude a better agreement between model and data:

(i)     on the spatial scale, our simulated concentrations are probably impacted by the very coarse resolution of the ocean model, which is critical to resolve the coastal ocean where a number of in-situ measurements of CO concentrations have been performed.

(ii)    On the temporal scale, the model does not explicitly resolve the diurnal cycle although the variations of surface





CO concentrations during the day have been shown to be first-order variations (Carpenter et al., 2012). As well, the model does not consider the inter-annual variability of ocean physics and biogeochemistry, as it is forced by a climatology of surface fields (temperature, winds, precipitation, …) (Aumont et al., 2015). This might be limiting for the evaluation of the simulated CO concentrations against *in situ* measurements, as it has not been always possible to average the data in order to be representative of daily means, and as the data are covering 50 years of measurement during which atmospheric forcing might have changed.

(iii)     In addition, we note a rather clear underestimation of the simulated CO concentrations when compared to in-situ measurements at high latitudes. This may result from a lower bacterial consumption rate in these regions or/and from higher CDOM levels associated to the presence of sea-ice or to riverine input of organic matter. Neither the supply of CDOM by sea ice nor by rivers are represented in the model, which may prevent an accurate representation of the CO cycle in those regions.

(iv)     Finally, our model does not include any representation of the so-called "dark production", a concept that had been first proposed by Kettle (2005) to explain high CO concentrations measured at depth. This term could be non-negligible as Zhang et al. (2008) estimated its contribution to be between 5 and 16 Tg C yr$^{-1}$. However, note that first this concept remains currently unclear and should be investigated in further laboratory experiments in order to provide modelers some potential parameterizations. Second, the analyses of the collected vertical profiles did not seem to clearly support the importance of such a mechanism to explain the differences with our simulated profiles.

We also want to emphasize the large uncertainties associated to the photoproduction term. Indeed, we have shown that the global photoproduction can vary within a factor of 3 (14.2-49.1 Tg C yr$^{-1}$) depending on the CDOM parameterization chosen. Hence, efforts should be focused toward a better understanding of the complex CDOM nature and on its representation in models. Especially as a good knowledge of its spatio-temporal distribution is critical not only for the CO production but also for a better understanding of remineralization of dissolved organic carbon in the ocean (Mopper and Kieber, 2001) and more generally for the biogeochemical cycles that are driven by light availability in the ocean (Shanmugam, 2011).

Finally, we compared our estimates of ocean CO emissions to that published by Erickson in 1989. Our emissions are quantitatively much closer from the most recent estimates and in better agreement in term of spatial distribution with the known processes controlling the oceanic CO. Future works will assess the impact of our estimates of CO emissions on the oxidizing capacity of the atmosphere with respect to that obtained with the canonical Erickson's estimates. Besides, the implementation of our ocean CO cycle module within an Earth system model will enable to explore potential Earth system feedbacks associated to the oceanic CO cycle.



**Acknowledgments**

We would like to thank the LEFE FORCAGE and EU-H2020-CRESCENDO (grant agreement No 641816) projects for funding. We also gratefully acknowledge Valérie Gros and Bernard Bonsang for their contribution and access to CO data from their TRANSSIZ and OOMPH cruises. A first version of the work was performed during the internship of Aïda

Bellataf in 2012 with the help of Sophie Tran. The authors are thankful to them and have a sad though to Aïda, a very dynamic and enthusiastic student who brutally died in 2017 at the age of 32. Finally, we thank the GlobColour project for providing access to the CDM content (http://globcolour.info) and the ASTM for the distribution of the ground-based solar spectral irradiance SMARTS2 (version 2.9.2) *Simple Model for Atmospheric Transmission of Sunshine* (https://www.astm.org/Standards/G173.htm).

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




**Table 1: photoproduction, direct phytoplankton production, bacterial sink and CO flux to the atmosphere at the global scale (TgC yr$^{-1}$) for the different simulations performed. For each experiment the oceanic CO inventory (TgC) and the RMSE value associated to the comparison with *in situ* CO concentrations are shown (nmol L$^{-1}$).**

| Experiment | CDOM parameterization | Consumption rate $k_{co}$ (d$^{-1}$) | Photoproduction (TgC yr$^{-1}$) | Direct phytoplankton production (TgC yr$^{-1}$) | Bacterial sink (TgC yr$^{-1}$) | CO flux (TgC yr$^{-1}$) | CO inventory (TgC) | RMSE (nmol L$^{-1}$) |
|---|---|---|---|---|---|---|---|---|
| **Standard** | **Morel, 2009** | **0.2** | **19.2** | **6.2** | **21.9** | **3.6** | **0.3** | **1.84** |
| Tests on the CDOM absorption coefficient | Launois et al., 2015 | 0.2 | 49.1 | 6.2 | 46.5 | 9.0 | 0.6 | 2.64 |
| | Preiswerk and Najjar, 2000 | 0.2 | 14.2 | 6.2 | 17.9 | 2.7 | 0.2 | 1.98 |
| Tests on the bacterial consumption rate | Morel, 2009 | 0.1 | 19.2 | 6.2 | 19.4 | 5.6 | 0.5 | 1.93 |
| | Morel, 2009 | 0.4 | 19.2 | 6.2 | 23.4 | 2.1 | 0.2 | 2.03 |
| | Morel, 2009 | 1.0 | 19.2 | 6.2 | 24.7 | 0.9 | 0.1 | 2.33 |
| | Morel, 2009 | 2.0 | 19.2 | 6.2 | 25.4 | 0.4 | 0.0 | 2.48 |
| | Morel, 2009 | Variable [mean=0.2] | 19.2 | 6.2 | 22.0 | 3.5 | 0.3 | 1.81 |

**Table 2: description of the observation datasets used to build the synthetic dataset of surface seawater CO concentrations.**

| Dataset | Sampling Location | Season/year | Sampling depth (when specified) | Type of data provided in the paper and digitalized | Data Treatment/ Remarks | Representativity of each point | Number of points |
|---|---|---|---|---|---|---|---|
| Bates et al. 1995 | Pacific Ocean 5 North-South transects | Different seasons between 1987 & 1994 | 5±5m | Daily means plotted in their Figure 3 | | Daily mean along path of 2-3° northward | 150 |
| Conrad et al. 1982 | Equatorial Atlantic Ocean | February 1979 | 1-4m depth | Fig 13 plot of mean diurnal profile of measurements | Average of the diurnal profile to obtain daily mean | 18-day mean over a 5° N-S transect | 1 |
| Jones 1991 | Sargasso Sea | June and Sept 1986 | | Day and night punctual values from their Fig 2C and 3C 5 obs. for 2 days in June 12 obs. for 3 days in Sept | Mean over the whole cruise | Mean for 2 or 3 days Fixed location | 2 |
| Kitidis et al. 2011 | Mauritanian upwelling filament | April-May 2009 | | Mean before noon & mean after noon given in the text p127 | | Mean of 14 punctual obs. the same day along path ~2° | 1 |



| OOMPH1 Gros & Bonsang, personal communication | South Atlantic E-W transect | January-February 2007 | | High frequency data provided directly by B. Bonsang and V. Gros | High frequency data averaged to daily means | Daily mean from roughly 150 observations along path ~5° | 14 |
|---|---|---|---|---|---|---|---|
| OOMPH2 Gros & Bonsang, personal communication | South Atlantic E-W transect | February-March 2007 | | High frequency data provided directly by B. Bonsang and V. Gros | High frequency data averaged to daily means | Daily mean from roughly 150 observations along path ~5° | 15 |
| Otha 1997 | Equatorial Pacific Upwelling region | Nov 1993 | | Data in Table 1 3 Daily Means | | Mean for 3 days Fixed location | 1 |
| Stubbins et al. 2006b | Atlantic Meridional Transect | April 2000 | | Daily min from fig 3 and daily max from Fig4 | Assumption Mean = (max-min)/2 | Pseudo Daily mean along path ~5° | 16 |
| Swinnerton & Lamontagne 1973 | South Pacific Ocean | November - December 1972 | | Daily Means from Fig 3 Identification of sites surrounded by ice | Flag for sampling sites surrounded by ice | Daily mean along path ~6-7° | 23 no ice 3 with ice |
| Swinnerton et al. 1970 | Tropical Atlantic Ocean | April 1969 | | Fig 1: 4 to 9 instantaneous values per day | Daily mean | Daily mean at fixed station | 2 |
| Xie et al. 2009 | South-eastern Beaufort Sea | Autumn 2003 / Spring 2004 | | Means and standard deviation from Table 3 | | Spring= Mean over 18 stations (sampling at different hours in a 1.6°lat x 5.9°lon rectangle) Autumn= Mean over 19 stations (sampling at different hours in a 1.9°lat x 15.6°lon rectangle) | 2 |
| Yang et al. 2011 | East China Sea Yellow Spring Sea | April-may 2009 | | Punctual values | Mean when > 3 obs/day | Daily mean over a small path (<2°) | 15 |
| Tran et al. 2013 | Arctic Ocean | June-July 2010 | Direct pumping of sea surface water by the ship | High frequency data provided directly by S. Tran Identification of sites surrounded by ice | Flag for sampling sites surrounded by ice | Daily Mean (between 46 and 246 obs/day) | 27 no ice 11 with ice |
| Zafiriou et al. 2008 | Sargasso Sea | August 1999 March 2000 | | Table 1 Mean of hourly obs for 9 days and for 12 days respectively | | Mean for a few days Fixed station | 2 |




**Table 3: description of the observation datasets used to build the synthetic dataset of vertical profiles of seawater CO concentrations**

| | Sampling Location | Season/year | Type of data provided in the paper and digitalized | Data Treatment/ Remarks | Representativity of each point | Number of profiles originally / in this dataset |
|---|---|---|---|---|---|---|
| Conrad et al. 1982 | Equatorial Atlantic Ocean 3°N to 2°S/22°W | February 1979 | 0-100meters, several hours from Fig 9 | Average of the 17 profiles to obtain an averaged profile | Mean over a 5° N-S transect | 17 / 1 |
| Day & Faloona 2009 | Northern California coastal upwelling System 38.2:38:3N 123:123.2W | February to May 2006 | 0-60m 4 individual profiles from Fig 3 for day and night | Coastal | Daily Mean for each site | 4 / 2 |
| Jones 1991 | Sargasso Sea 27°N, 73°W | June 1986, September 1986 and June 1987 | 6 individual profiles over 0-200m (fig 2, 3 4) & one mean profile over 0-4500m (fig 5) | | For 0-200m, interpolation on similar levels and average of the 6 profiles | 6 / 1 |
| Johnson & Bates 1996 | Tropical Pacific Ocean RITS93 19°S,149.5°W and RITS94 4°S,140°W to 4°34'Sto140°55'W | April 1993 for 19°S,149.W and December 1993 for 4°S,140.W | 0-250meters, for each site one profile at sunrise and 1 in the afternoon 2 contrasted sited sites : 1 static profile in oligotrophic area (RITS 93) & 1 in upwelling area (Lagrangian, ship, RITS 94) Fig 1 | Mean of Sunrise and Afternoon profiles | Mean of 2 profiles per day | 4 /2 |
| Otha 1997 | Equatorial Pacific Upwelling region 0°N-159°W | November 1993 | 0-100meters, several hours Data from fig 2 profiles at different hours (two days, same location) | Average | Mean for 2 days (3 prof/day) Fixed location | 6 / 1 |
| Swinnerton &Lamontagne 1973 | South Pacific Ocean 21N,77S/129W,177E | November - December 1972 | 0 – 100m vertical profiles at 14:00 local time from Fig 4 | not indicated | One location at 14:00 | 10/10 |
| Xie et al. 2009 | South-eastern Beaufort Sea spring 70.2:71.8°N 123.5:129.4°E & autumn 69.9:71.8°N 123.1:138.7°E | Autumn 2003 / Spring 2004 | 0 – 50m vertical profiles Means and standard deviation from Fig 3 | | Spring= Mean of 37 profiles from 14 stations Autumn= Mean over 16 profiles from 14 stations | 2 |
| Yang et al. 2011 | East China Sea Yellow Spring Sea 25.4-26.9°N 120.5-122.8°E & 30.8N:128.3E & 27.6N:126.2E | April-May 2009 | 0-900m, several hours (Fig 3 and 4) | | 8 of ten are averaged because on the same transect. One other is left alone. | 10 / 2 |
| Tran et al. 2013 | Arctic Ocean 75°N, -7W:+8E & 23 profiles at 78.5°N -12W:+8E | June-July 2010 | 0-100m 29 profiles provided directly by S. Tran | Average of 29 profiles | | 29/1 |
| Zafiriou et al. 2008 | Sargasso Sea August 31.37:31.50°N 64.00:64.03°W March 32.01:32.03°N 64.01:64.02°W | August 1999 and March 2000 | 0-100m Fig 9 provides 3 mean profiles for different periods of the day (each based 7 or 8 profiles) | | Averaged in 1 profile per period (march or august), can be considered as representative of daily means for 7 days | 15 / 2 |





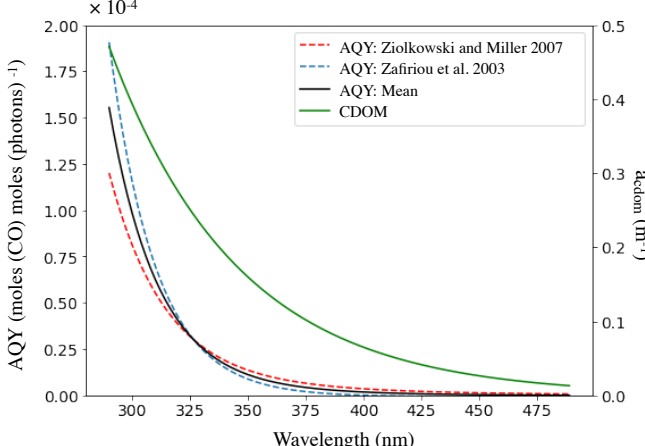

**Figure 1: CO apparent quantum yield (AQY in moles of CO per moles of photons) and CDOM absorption coefficient ($a_{cdom}$ in m$^{-1}$) as a function of the wavelength (nm). For the AQY, the parameterizations of Ziolkowski and Miller (2007) and Zafiriou et al.**
5 **(2003) are shown with dotted lines and the resulting mean relation used in PISCES is shown with a continuous line. For $a_{cdom}$, the relation is shown for a Chla concentration of 1 mg m$^{-3}$ and is calculated according to the parameterization of Morel (2009).**

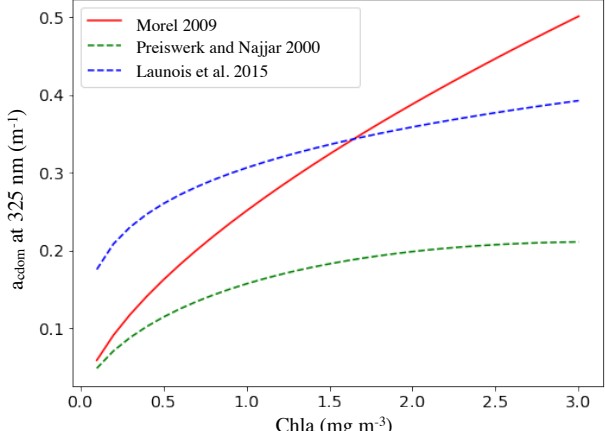

**Figure 2: CDOM absorption coefficient (m$^{-1}$) at 325 nm as a function of the Chla concentration (mg m$^{-3}$) for the parameterizations of Morel (2009), Preiswerk and Najjar (2000) and Launois et al. (2015). The continuous line indicates the chosen parameterization.**



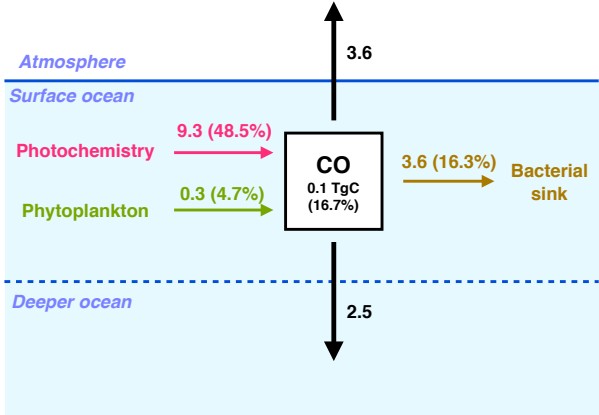

**Figure 3: global fluxes in the surface ocean (between 0 and 10 meters) of the oceanic CO sources and sinks (in TgC yr⁻¹). For each biological term (photoproduction, direct phytoplankton production and bacterial consumption), the relative contribution of the surface layer to the whole water column budget is shown as a percentage.**

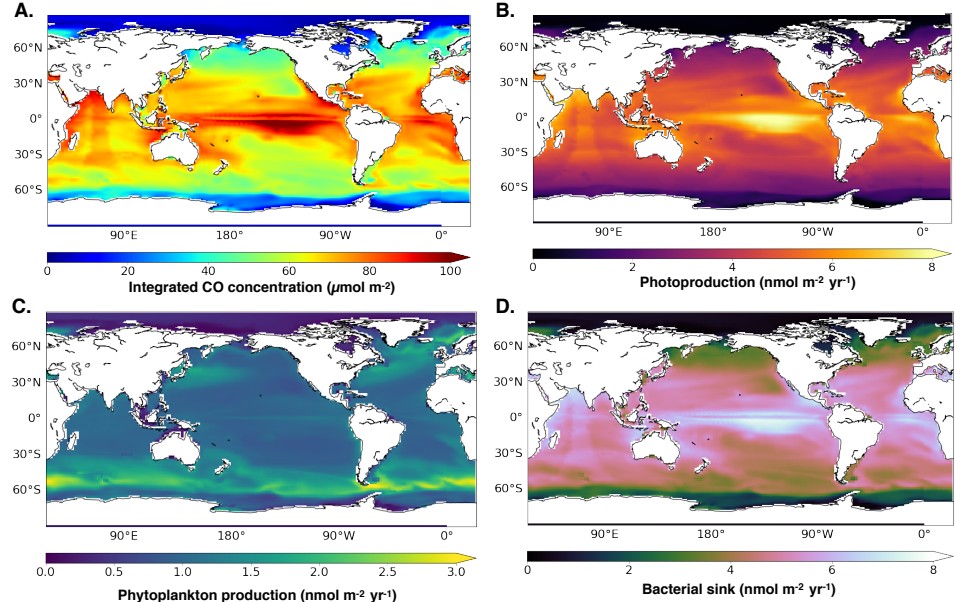

**Figure 4: spatial distribution of oceanic CO concentration (A.), photoproduction (B.), direct phytoplankton production (C.) and**

10     **bacterial sink (D.), vertically integrated upon 1000 meters.**




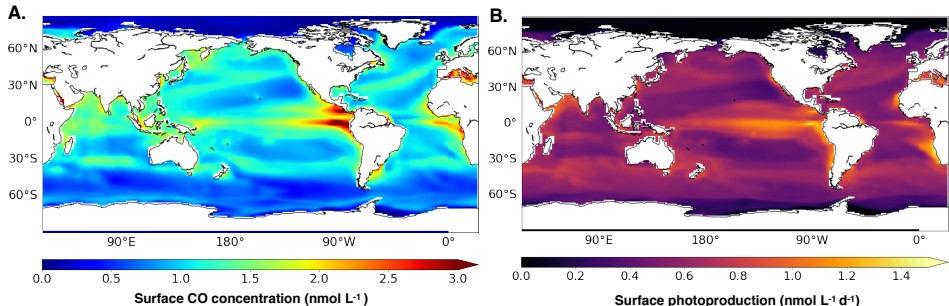

**Figure 5: spatial distribution of oceanic CO concentration (A.) and photoproduction (B.) between 0 and 10 meters.**

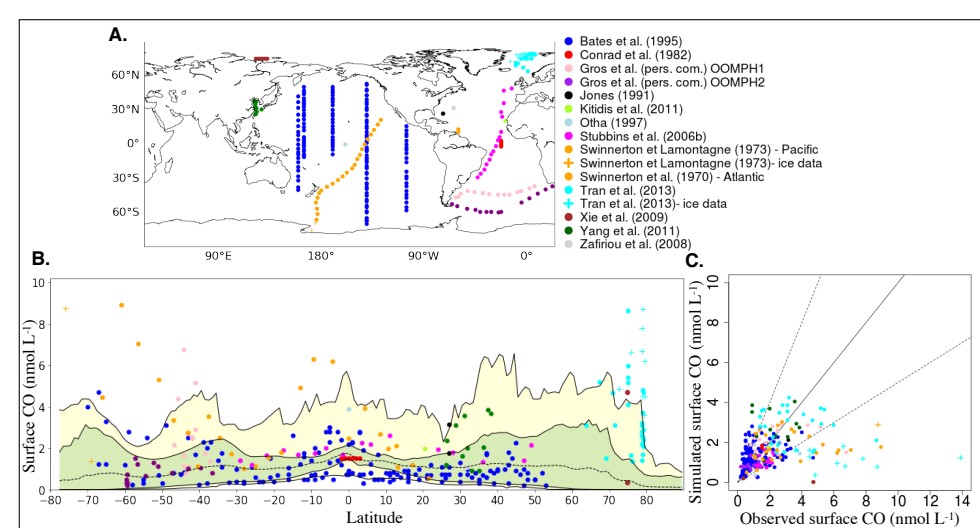

**Figure 6: comparison of** *in situ* **oceanic CO concentrations between 0 and 10 meters simulated with PISCES with the ones measured in the surface ocean. A: position of the surface measurements collected in the literature. B: Surface CO concentrations as a function of the latitude [Dots = observed concentrations; dotted line = longitudinal and monthly mean simulated CO concentration; green area = interval between maximum and minimum longitudinal mean concentrations; yellow area = interval between maximum and minimum CO concentrations at a given latitude]. Remark= A data point of 13.9 nmol L⁻¹ from the dataset of Tran et al. (2013) has been removed for better visibility. C: Scatter plot the of the simulated CO surface concentrations vs the observed ones. The solid line represents the 1-1 line and dotted lines the 2-1 and 2-1 lines.**



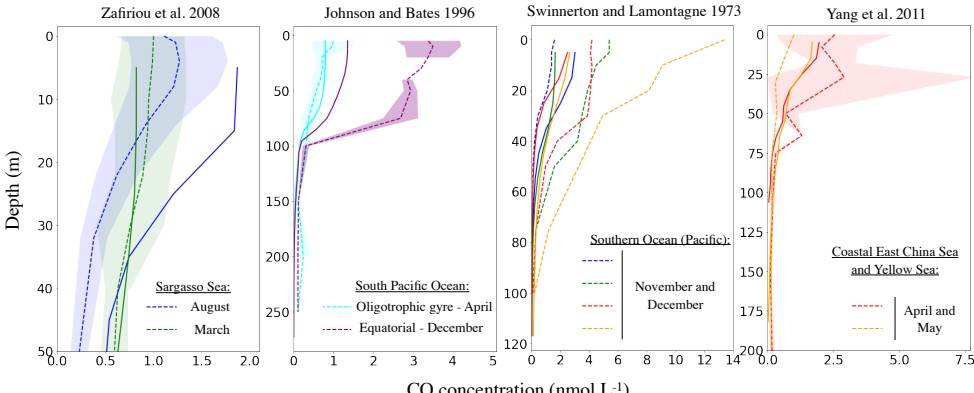

**Figure 7: comparison of simulated oceanic CO profiles with the measured ones (simulated profiles for the same months and location than measurement). Only a few datasets are shown in this figure (others are available in the Supplement). Dotted lines**
5 **represent the observed profiles and continuous lines the model output. The standard deviation of the observation, when available, is shown around each observed profiles with the shaded area of the same color.**

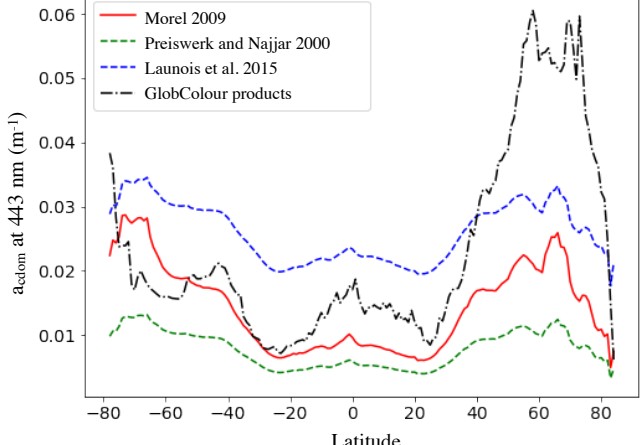

**Figure 8: mean CDOM absorption coefficient at 443 nm (m⁻¹) as a function of the latitude, for the different parameterizations tested. The mean surface absorption coefficients for the colored detrital matter, retrieved from GlobColour products at 443 nm**
15 **and averaged over 2002-2012, are also shown.**




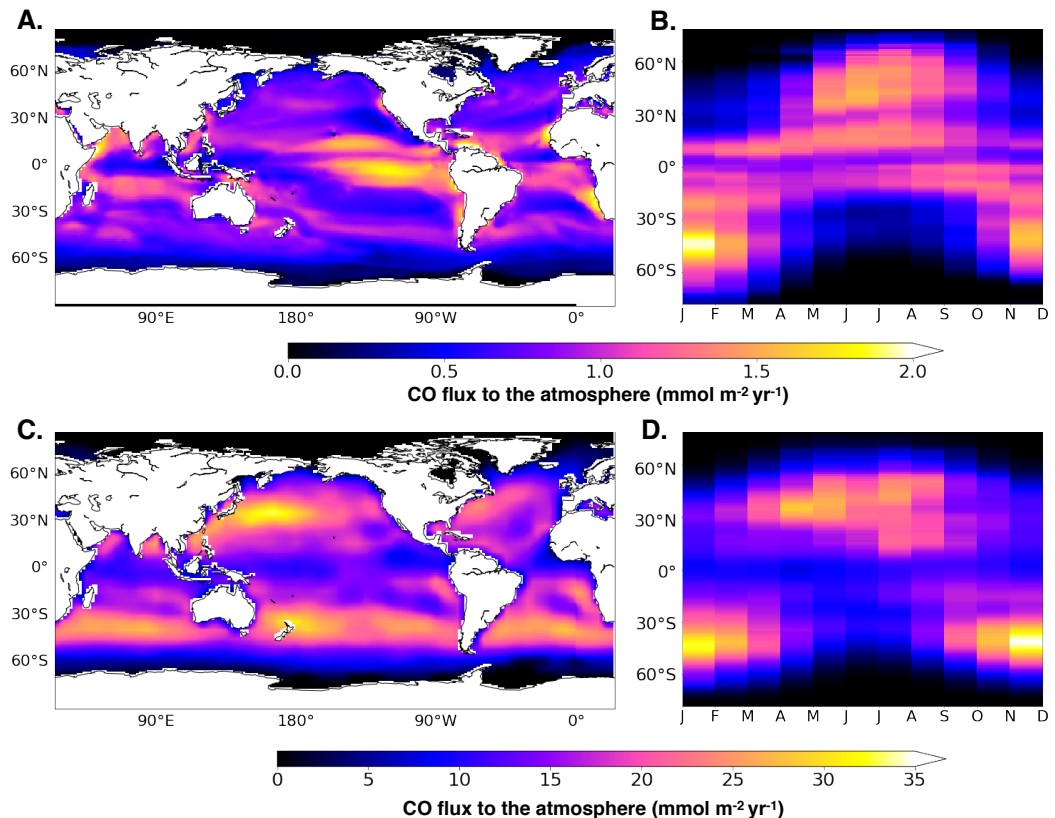

**Figure 9: oceanic CO emissions simulated by PISCES (A and B) or by the model of Erickson (1989) (C and D). Panels A and C**
5  **present the spatial distribution of annual mean emissions. Panels B and D present the mean seasonal variation with latitude. All**
**fluxes are in mmol m$^{-2}$ yr$^{-1}$ and are directed toward the atmosphere.**





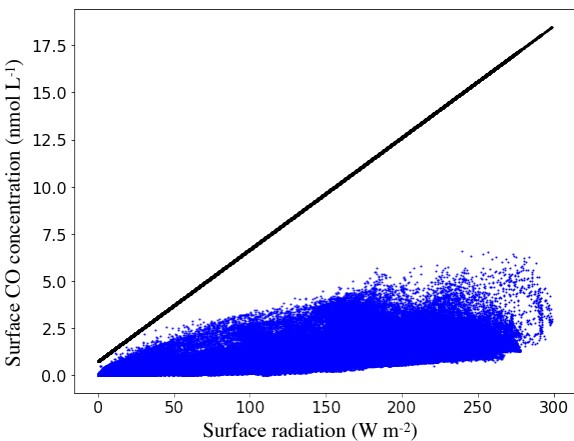

**Figure 10: surface CO concentrations as a function of the total solar radiation available at the surface ocean. Blue dots are mean annual surface concentrations retrieved from PISCES and the black line is the linear relation used by Erickson (1989).**

