# Peer review of "The oceanic cycle of carbon monoxide and its emissions to the atmosphere"

_Biogeosciences, 2018_

## Referee Comment (RC1) · Anonymous Referee #1 · 16 Oct 2018

Review of 'The oceanic cycle of carbon monoxide and its emissions to the atmosphere', ms by Conte et al. submitted to Biogeosci Discuss, ms# bg-2018-410.

General comments Carbon monoxide (CO) is an important atmospheric trace gas. The oceans (incl. open ocean and coastal oceans) are sources of atmospheric CO, however measurements of diss. CO are analyticaly challenging and thus only a few data sets have been published so far. To this end it is not a big surprise that its consumption and production pathways in the ocean are not well known and the oceanic CO emission estimates are associated with a very large degree of uncertainty. The ms under review presents the results of a new modelling study of the oceanic CO pathways and its emissions to the atmosphere. Despite the fact that the model approach and the chosen input parameters are reasonable, I have some major concerns (see below) which

[Figure]

should be carefully addressed in a revised version.

Major comments

- Coastal oceans (incl shelf areas, estuaries etc) are important sources of CDOM which, in turn, is the prerequisite of the photochemical production of CO. I am wondering why the role of coastal oceans is not discussed in the article. It is only mentioned briefly on page 17, lines 30-33. I understand that the model is not suitable to simulate coastal oceans (shelf areas, estuaries etc.). To this end, the authors should modify the ms title and the discussion by stating that their results are only valid for the open ocean or include a discussion of CO in coastal areas (i.e. contribution to CO emissions etc).

- Important literature has been ignored:

Kawagucci, S., et al. (2014). "Molecular hydrogen and carbon monoxide in seawater in an area adjacent to Kuroshio and Honshu Island in Japan." Mar. Chem. 164: 75-83.

Park, K. and T. S. Rhee (2016). "Oceanic source strength of carbon monoxide on the basis of basin-wide observations in the Atlantic." Environmental Science-Processes & Impacts 18(1): 104-114.

Xie, H. X. and O. C. Zafiriou (2009). "Evidence for significant photochemical production of carbon monoxide by particles in coastal and oligotrophic marine waters." Geophys. Res. Lett. 36.

Yang, G. P., et al. (2010). "Distribution, flux and biological consumption of carbon monoxide in the Southern Yellow Sea and the East China Sea." Mar. Chem. 122(1-4): 74-82.

- The dark production (DP), which was shown by Zhang et al., (2008)to be a significant additional source of CO, has been ignored in the model approach (see equation (1)). However, in the conclusions (page 18, line 12-18) it is stated that '[. . .] analyses of the collected vertical profiles did not seem to clearly support the importance of such a mechanism to explain the differences with our simulated profiles.' This is too vague

and not acceptable. I think that the correct scientific approach to tackle this 'problem' would be to include the DP (I guess you can use the parameterization given by Zhang et al., 2008) in equation (1) and show the results of model runs with DP/without DP. Only based on these model results you will be able to assess the role of the DP.

Minor points - Page 3, line 21: please give the correct chemical formulas for nitrate, ammonium, phosphate, and iron.

- Page 6, section 2.1.4: please note that fCO is a '(dry) mole fraction' (it is not correct to call it a 'mixing ratio' or a 'concentration').

- Page 6, line 19: In view of the pronounced spatial and temporal variability of atm CO I am wondering why the atm CO was set to fixed global mean. Please discuss.

- Page 7, wind speed: Please state whether you used a global mean wind speed (which value? ref?) or whether a global wind field (ref?) was used for the computation of the air/sea gas exchange. -

---

## Referee Comment (RC2) · Anonymous Referee #2 · 18 Oct 2018

General comments

In their study the Conte et al present a novel modeling scheme for marine carbon monoxide source and sink processes implemented into a state-of the art ocean biogeochemistry model. They successfully evaluate the model against available observations and present important novel insights into the spatial distribution of open ocean emissions of CO. This work will be useful to readers of BG due to the relevance of CO for a number of atmospheric chemical cycles, including tropospheric chemistry of OH, CH4, O3. I recomment publishing the paper after addressing the minor revisions below that primarily aim at helping to improve the presentation of results and methods used manuscript.

Specific comments

[Figure]

The short lifetime of CO in water implies large spatial and temporal variability. Related to this I have a few questions/ comments:

• If I get it correctly, for simulating CO you decided to choose a model setup that uses climatological forcing that does not resolve diurnal light cycle. Forcing data like NCEP or ERA interim however do resolve SW radiation fluxes on a 6 or 3 hourly basis. Given that you are using a sophisticated wave-length dependent CO production mechanism omitting on the other hand this feature seems like an odd decision on complexity versus simplification. As production and decay rates are highly uncertain, does resolving wave length dependency and omitting the day night cycle ( thus also dark production, as described in Day and Faloona 2009, JGR Oceans 114) imply you expect the former to propagation stronger into the presented mean solution than the latter, i.e. more important to resolve in a realistic CO production scheme? • How does the seasonal cycle of CO look like in the model? Is the quality of the model solution different for different seasons, i.e. how does it relate to the models' ability to represent the seasonal cycle of Chl-a? • In the evaluation of the concentrations you a using model data collocated to observational data. Does this mean you are using individual grid cells? If so, are these representative for a larger surrounding area – did you consider averaging several grid cells, as physical features such as the extent of subtropical gyres, location of fronts etc are not geo-referenced, i.e. collocated with real world conditions during the ship cruises? In particular for the vertical profiles it could be useful instead of showing one profile adding its variability taking into account several neighboring cells (and eventually temporal standard deviations for within the averaging period).

Given that the compilation of observational data is presented to be unique and its averaging methods are very diverse I suggest to extend the section observations. E.g. it would be interesting to learn, also in light of the large temporal variability of CO concentrations, if certain months/seasons are better resolved in the observations than others. Now I can deduce this only from the tables, but do not get any direct information in the main text. Also, it would be important to know on how many values per averaging

period the temporal means are based, as e.g. the observed diurnal cycle is very strong and party not symmetric (afternoon maxima). Furthermore, I am do not know if CO measurement techniques are comparable across the different observational sources wrt the limit of quantification/ detection, or if there was development in the methodology from the 1970s to now. Are all of the published observational data equally reliable, in particular wrt to low CO values?

I struggled with the structure of the manuscript, presenting first simulation results and evaluation of the "standard" experiments followed by a separate discussion of the sensitivity experiments.

Statements as " indicating again a possible bias in the production process " in the section on the standard experiment could be easily complemented by the results of the sensitivity studies on process parametrization, instead of having to collect this information later in the manuscript. Also this separation of the evaluation of surface concentrations leads to inconsistent level of discussion of potential sources of discrepancies: Whereas in the standard experiment it was argued that missing processes related to sea ice or a missing spatial variability of the decomposition rate might be causes of discrepancies in polar regions, only later in the section of CO production the authors state that also missing terrestrial CDOM sources might be a source of model data discrepancies. It is not clear to me how it was decided which of the parameterizations are chosen to be "standard" vs "alternative". For example, I understand that choosing the Launois et al. 2015 CDOM parameterization leads to high CO production and in combination with a consumption rate of 0.2 d-1 CO concentrations get too high compared to observations. On the other hand using the "standard" parameterization together with a consumption rate of 1 d-1 lead to very low CO concentrations. The combination of the Launois et al. 2015 CDOM parameterization and a fast consumption rate was however not tested or presented without commenting on the reasons for this. If instead the authors would present all of the tested parameterizations in the methods part equally, present first an evaluation of the model results wrt to the range of parameters chosen

and process parameterizations, and resulting from this discuss the source and sink budget, and emissions only for the most successful configuration, decisions taken and it's consequences could become clearer.

The model evaluation is lacking discussion of the simulated physical ocean solution, e.g. whereas contributions of MLD are mentioned in possible causes of discrepancies of modelled and observed CO concentrations these are not compared to the ship cruise or climatological T,S, or MLD data. I guess it would be possible to get the CTD data of the ship cruises and compare them. It would be useful to see in particular in the analysis of the vertical profiles, but also for the surface data whether how NEMO performs in regions with CO data. Furthermore the discussion of the evaluation could be more detailed in discussion the quality of the Chl-a solution, which influences CO production.

The authors assume a constant homogeneous atmospheric mixing ratio of CO in their emission calculation. As major sources of CO (fossil fuel combustion, biomass burning) are on land and a major sink is reaction with of a large hemispheric and seasonal variation of CO mixing ratios in air due to the continental distribution and OH seasonality is expected. Is the over-saturation of the ocean indeed that strong that these variations can be omitted in the emission calculation?

p8l23 .. same forcing fields as the ones in Aumont ea 2015.. Please help the reader to easily understand implications of your model setup into the results by repeating main characteristics of that forcing (source, spatial and temporal resolution). p10l10: .. all vertically integrated over the upper 1000m. The vertical profiles suggest that below the euphotic zone there is not much CO left, why do you choose to integrate over 1000m?

P12l24 .. which can be related to differences in the light penetration and mixed layer depth. Please be more specific.. is the simulated MLD generally too low/high? Do you indicate a different mixing scheme would improve the profiles? How does the model's vertical resolution in the upper ocean affect the vertical profiles?

P13 l17ff: .. those minimums are best represented by the relation of Morel since the ones in Launois .. give too high and too low values. Could you please comment on the simulated Chl field here, so that it get clear that these CDOM parameterizations are responsible for the discrepancies rather than the simulated Chl. Furthermore, satellite derived observations are based on a number of assumptions (e.g. also wrt to light penetration depth in turbid and non-turbid waters) and models (bio-optical, atmospheric correction..), in particular in derived product as CDM.

Conclusions ii) .. the model does not consider the inter annual variability of ocean physics and biogeochemistry.. Even with a climatological forcing both ocean physics and biogeochemistry solutions will show inter annual variability due to e.g. fluid dynamics (wave propagation) and different plankton over-wintering stocks. Clearly this variability will be lower than in simulations using transient forced boundary conditions or coupled atmosphere-ocean simulations. I suggest to rephrase this statement slightly.

Technical correction

Fig.7 : I find the color coding of the lines confusing.. is it time or location, what does it show in the subplot 'Swinnerton and Lamontagne 1973' ?

[Figure]

---

## Short Comment (SC1) · 5 Nov 2018

*Note to the editor and authors: As part of an introductory course to the Master programme Earth & Environment at Wageningen University, students get the assignment to review a scientific paper. Since several years, students have been reviewing papers that are in open online discussion for Copernicus Journals, and the top students in the class have been asked to submit their reports to the discussion in order to help the review process. While these reports are written in the form of official (invited) reviews, they were not requested for by the editor, and we leave it up to the editor and authors to use these reports to their advantage. We hope that these reports will positively contribute to the scientific discussion and to the quality of papers published. This report/review was supervised by Prof. Wouter Peters.*

**Review on "The Oceanic cycle of carbon monoxide  and its emissions to the atmosphere" by Conte *et al.***

*Summary and Scope*

Carbon monoxide (CO) is important in the atmospheric chemistry, because it affects the lifetime of greenhouse gasses, it is a sink for hydroxyl radicals and it affects the ozone chemistry. The ocean is one of the sources of CO. Although it is a minor source, it can be important far away from anthropogenic sources. However, the oceanic CO source is still poorly estimated, mainly because the main driver, biological activity, has a large spatial and temporal variability. This paper aims to overcome this lack of knowledge by using a biochemical model called PISCES (Pelagic Interaction Scheme for Carbon and Ecosystem studies) to assess the CO dynamics at the global scale. It explicitly looks at the interaction between the ocean and the atmosphere. Therefore, they added an extra module to represent the CO sources and sinks. They compared the model output to a gathered dataset of *in situ* CO measurements of the last 50 years. This is the first time a compilation was made of various *in situ* CO measurements. The combination of using a new modified model and this gathered data set, to better estimate the global oceanic CO source, makes this research in particular novel. Moreover, the paper also included the direct production of CO by phytoplankton in their research, which is recently be reported as a CO source and only measured once by Gros *et al.* in 2009. Therefore, this paper makes use of new knowledge in their research field.

The paper is well written and a clear structure of the different parts of the research is given. The thorough introduction nicely introduces the topic and describes why it is particularly relevant, which gives the impression that the authors know where they are talking about. In addition, the authors are well aware of the limitations of the used model, which they nicely point out in their conclusion.

The methodology is explained well and is similar to the methods used in other studies in this research field. Stubbins *et al.* (2006), for instance, also multiplied the apparent quantum yield with the moles of photons absorbed (solar irradiance) to calculate the CO photoproduction term. A bit broader spectrum was used though: 280-800 nm. Whereas this paper uses mainly earlier found empirical relationships, Stubbins *et al.* (2006) measured it mainly themselves. They also compared those to

empirical relations, similar to the ones used in the review paper. Hence, although there are some differences in the way data was gathered, the approach to calculate photoproduction is similar.

In general, the paper nicely builds on former research and covers different domains of the earth system with a main focus on the atmosphere, ocean and biology. Therefore, it nicely fits the scope of the journal. For instance, atmospheric models can use the new data of oceanic CO emissions to investigate the oxidizing capacity of the atmosphere in remote oceans. This will replace the outdated model estimation of the global oceanic CO emissions by Erickson (1989), who used a simple empirical relationship between solar radiation and the CO concentration. However, there are a couple of things which need to be improved before the paper can be accepted.

**Major arguments**

Although the overall methodology is clear to me, there is one main point I would like to be discussed more thoroughly. In the section "Spectral solar irradiance" (p. 4), they use attenuation coefficients from earlier conducted research, to determine the penetration depth of irradiance. The data was not available for all wavelengths in the range used in this paper. Therefore, the authors applied linear interpolation to the remaining wavelengths to retrieve the coefficients. However, I wonder whether they really vary linearly. If not, this may chance the outcome of formula 3 in line 17 of the "Spectral solar irradiance" section (p. 4), which, in turn, affects the outcome of formula 10 in line 17 of the "Photoproduction" section (p. 5). Hence, this could give rise to significant errors in the CO photoproduction term. This will directly affect the values given for photoproduction term in figures 3-5 (p. 28-29). Because the photoproduction term is one of the main sources of CO, a different value of the photoproduction term also chances the CO concentration. Therefore, figure 6 and 7 (p. 30-31), which show the modeled CO concentration, are indirectly affected too.

To make the choice of linear interpolation more straightforward, a figure should be made as figure 1 and 2, showing three coefficients as a function of wavelength (p. 28). In this graph the (linear) trendline with formula and the $R^2$ should be shown to see how strong the (linear) relationship is of the different coefficients with wavelength. This can be put in the "Spectral solar irradiance section" (p. 4). Furthermore, I would like to see at least a short discussion on the effect of linear interpolation on the CO photoproduction term, preferably under the section "3.3.1. CO photoproduction" (p. 13).

A second concern is the conclusion which states: "the model is able to predict most of the ~300 *In situ* data points within a factor 2. However, this means that it is possible that the model can over or underestimates a concentration by 100%! Hence, this is an extremely weak formulation, since this states that the model actually is not performing that well. Before further consideration of this script, I need to see an alternative metric for model performance implemented, which will be used to thoroughly discuss the prediction offsets of the model to the in-situ measurements, especially the large offsets. The RMSE error is a possible alternative metric, which is already calculated in the paper to test different parameterizations (p. 9 in the "2.4 Comparison to in situ" section). It is, however, not used in the main conclusion about the model performance. Besides that, such metric should not only be calculated for the entire sample of *in situ* measurements (285 subsamples), but also calculated for the subdivisions, in other words latitude groups representing different climate zones. This can be put in a new table and the results can subsequently be discussed in section 3.2.1. Surface CO concentrations (p. 11-12). Lines 5 and 15, which state "within a factor of 2", should then be edited in this section. Also,

figure 6C should then be replaced by a figure showing the offsets of the model simulation from the observed data (residuals).

Finally, the paper shows in figure 6 (p. 28) the locations of the different *in situ* measurements and how well the model predicts the CO concentration of these points for a given latitude. This is definitely an important figure. However, it could certainly be improved, since it contains a lot of information, and is hard to read and interpret. Especially 6B is messy: latitude on the x axis is shown in a different way than in the figure 6A (shows the locations of the *in situ* measurements). Also, shaded areas of mean minimum and maximum CO concentration are shown for latitude and longitude. I would recommend to leave the shaded area of the longitude out, since it does not tell more than the latitude averages and makes it even harder to read. Actually, I do not know what the longitudinal mean stands for in that graph, since for a particular latitude you have only one point for every longitude. Hence, it is the mean minima and maxima of one point. Only if it is of real importance for this research and it can be clearly motivated, it can be kept in.

In addition, the section that describes the figure uses a lot of information, which is hard to obtain from this figure. To make the graph more readable, figure 6A should be split from 6B and 6C and put it in a separate figure. Moreover, the colors and symbols of the different types of *in situ* data need to be adjusted. Now, the *in situ* data are hard to distinguish from each other, especially the ones with a dark color. I would like to see the dots slightly bigger and a clear distinct color set should be chosen.

***Minor arguments***

Besides these three main concerns, there are some minor issues and typos that have to be revised:

- Minor issue 1: the explanation of the used PISCES model is very short. I like the chosen model, but I wonder why in particular the PISCES model is chosen. Is it because the authors are most familiar with this model or is it because it simulates the phytoplankton concentration best? Secondly, I do not know what the type and resolution (both temporal as spatial) of the in- and output data are. This can be important, since this determines what conclusion can be drawn. Please, provide a longer description of the model including the above-mentioned information in the section 2.1 Oceanic CO model description.

- Minor issue 2: in the section "Efficiency of the excited CDOM to produce CO", the average of two parametrization is used to calculate the apparent quantum yield. Here, a short motivation should be provided.

- Minor issue 3: in section "3.2.1. Spatial patterns of the sources and sinks", it is mentioned that the annual mean of the CO concentration is vertically integrated over a depth of 1000 meters. Please provide a short motivation, why in particular a depth of 1000 meters is chosen.

- Minor issue 4: in the section "Sensitivity of the oceanic CO budget to changes in the photoproduction", the parameterization test is discussed. It is concluded that the Launois et al. showed the best results compared to the standard one of Morel. However, no reason was

given why still the standard parametrization of Morel is used. An argument should be given in this section.

- P. 3, line 5: a comma should be placed after "Hence"

- P. 4, line 11: reference missing of the SMARTs2 model

- P. 12, line 27: "April" should be changed to "August".

- P. 16, line 33: replace "from" by "to"

- P. 17, line 6: a comma should be placed after "In figure 10" and "is shown" should be placed after "PISCES"

- P. 29, figure 4: nmol m$^{-2}$ yr$^{-1}$ as unit for the graphs of photoproduction, phytoplankton production and bacterial sink, whereas in section "3.1.2. Spatial patterns of the sources and sinks" the unit mmol m$^{-2}$ yr$^{-1}$. Hence, the units in figure 4 should be changed to mmol m$^{-2}$ yr$^{-1}$

- P.31, figure 7: the x axis values are with and without commas. This should be changed.

References:

- Erickson, D. J. (1989). Ocean to atmosphere carbon monoxide flux: Global inventory and climate implications. *Global Biogeochemical Cycles*, *3*(4), 305-314.

- Gros, V., Peeken, I., Bluhm, K., Zöllner, E., Sarda-Esteve, R., & Bonsang, B. (2009). Carbon monoxide emissions by phytoplankton: evidence from laboratory experiments. Environmental chemistry, 6(5), 369-379.

- Stubbins, A., Uher, G., Law, C. S., Mopper, K., Robinson, C., & Upstill-Goddard, R. C. (2006). Open-ocean carbon monoxide photoproduction. *Deep Sea Research Part II: Topical Studies in Oceanography*, *53*(14-16), 1695-1705.

---

## Author Comment (AC1) · 30 Nov 2018

**Response for Referee #1**

We thank the referee for his comments and suggestions. We will strive to address each specific concern in detail.

**1) Major comments**

**Comment 1 — Coastal oceans** (including shelf areas, estuaries etc) are important sources of CDOM which, in turn, is the prerequisite of the photochemical production of CO. I am wondering why the role of coastal oceans is not discussed in the article. It is only mentioned briefly on page 17, lines 30-33. I understand that the model is not suitable to simulate coastal oceans (shelf areas, estuaries etc.). To this end, the authors should modify the ms title and the discussion by stating that their results are only valid for the open ocean or include a discussion of CO in coastal areas (i.e. contribution to CO emissions etc).

**Reply:** we agree that coastal areas could be important for the oceanic CO cycle. In the next manuscript version, we propose to state in the title that our study mainly deals with the open ocean and to better discuss the implications of not resolving properly the coastal ocean in the discussion section. To be more precise, PISCES does include a crude representation of the coastal areas, as some specific processes are represented (riverine inputs, iron input by sediment re-suspension, or coastal upwellings). However, these areas are represented with large uncertainties, mainly due to the low horizontal resolution chosen. Indeed, a horizontal resolution of ~ 200 km does not allow to fully resolve some fine-scale coastal processes such as tides or mesoscale and submesoscale eddies and associated upwelling. Coastal bathymetry and complex coastal currents would be much better represented with the same model using much higher horizontal resolution (see Bourgeois et al. 2016 for a application of the global NEMO-PISCES model at higher horizontal resolution).

**Comment 2 — Important literature has been ignored:**

- Kawagucci, S., et al. (2014). "Molecular hydrogen and carbon monoxide in seawater in an area adjacent to Kuroshio and Honshu Island in Japan." Mar. Chem. 164: 75-83.

**Reply:** we will consider the vertical CO profiles as well as the surface concentrations measured by Kawagucci et al. in our next manuscript version.

- Park, K. and T. S. Rhee (2016). "Oceanic source strength of carbon monoxide on the basis of basin-wide observations in the Atlantic." Environmental Science-Processes & Impacts 18(1): 104-114.

**Reply:** we will consider the oceanic CO concentrations measured by Park and Rhee in our next manuscript version, as well as discuss our global CO emission estimate against their estimate (4-24 Tg CO $yr^{-1}$).

- Xie, H. X. and O. C. Zafiriou (2009). "Evidence for significant photochemical production of carbon monoxide by particles in coastal and oligotrophic marine waters." Geophys. Res. Lett. 36.

**Reply:** we thank you for pointing out this article. We will discuss the potential CO photoproduction by organic particles as the article suggests this process to be of importance for both coastal and blue waters.

- Yang, G. P., et al. (2010). "Distribution, flux and biological consumption of carbon monoxide in the Southern Yellow Sea and the East China Sea." Mar. Chem. 122(1-4): 74-82.

**Reply:** we did consider Yang et al. oceanic CO measurements. However, due to the coarse model resolution and the very coastal location of the data, we are not able to include these data in our evaluation as there is no model grid cell associated to the location of their measurements.

**Comment 3: The dark production** (DP), which was shown by Zhang et al. (2008) to be a significant additional source of CO, has been ignored in the model approach (see equation (1)). However, in the conclusions (page 18, line 12-18) it is stated that '[...] analyses of the collected vertical profiles did not seem to clearly support the importance of such a mechanism to explain the differences with our simulated profiles.' This is too vague and not acceptable. I think that the correct scientific approach to tackle this 'problem' would be to include the DP (I guess you can use the parameterization given by Zhang et al., 2008) in equation (1) and show the results of model runs with DP/without DP. Only based on these model results you will be able to assess the role of the DP.

**Reply:** we initially chose in our study to only represent the established sources and sinks of the oceanic CO, and the ones for which global or open ocean parameterizations exist. The dark production is an issue as the mechanism associated to this process is not yet totally established (consumption of the CDOM by heterotrophic process? or physicochemical process?). Furthermore,

there is no parameterization for the open ocean. Indeed, Zhang et al. (2008) developed one for absorption coefficients at 350 nm ($a_{350}$) of more than 0.23 m$^{-1}$. PISCES describes the global oceanic $a_{350}$ but with an annual mean in the surface ocean of 0.06 m$^{-1}$(standard deviation is 0.04 m$^{-1}$, with a minimum value of 0.01 m$^{-1}$ and a maximal value of 0.38 m$^{-1}$ reached in coastal areas). Therefore, the parameterization might not be suitable for a use in a global, blue water model. Zhang et al. themselves also suggest that extrapolating the parameterization to the open ocean may lead to large uncertainties. Considering that neglecting the dark production is highly questionable, we propose in our next manuscript version to better discuss this process and to test the parameterization of Zhang et al. (2008) in PISCES in order to estimate the error we make by neglecting it in our best guess simulation.

**2) **Minor comments:**

**Comment 1:** Page 3, line 21: please give the correct chemical formulas for nitrate, ammonium, phosphate, and iron.
**Reply:** Yes, $NO_3$, $NH_4$, $PO_4$, Si and Fe, will be changed for $NO_3^-$, $NH_4^+$, $PO_4^{3-}$, $Si(OH)_4$ and dissolved Fe.

**Comment 2:** Page 6, section 2.1.4: please note that fCO is a '(dry) mole fraction' (it is not correct to call it a 'mixing ratio' or a 'concentration').
**Reply:** We will replace the term 'mixing ratio' by 'dry mole fraction'.

**Comment 3:** Page 6, line 19: In view of the pronounced spatial and temporal variability of atm CO I am wondering why the atm CO was set to fixed global mean. Please discuss.
**Reply:** Tests had previously been performed about the atmospheric CO but were not shown in the manuscript version. They show that the value of the atmospheric CO dry mole fraction is of little influence on the oceanic emission. For example, using a homogeneous and constant CO dry mole fraction of 45 pptv leads to a global oceanic CO emission of 3.7 Tg C yr$^{-1}$ (against 3.6 Tg C yr$^{-1}$ using 90 pptv). We will specify these results in the next manuscript version.

**Comment 4:** Page 7, wind speed: Please state whether you used a global mean wind speed (which

value? ref?) or whether a global wind field (ref?) was used for the computation of the air/sea gas exchange.

**Reply:** As mentioned in Aumont et al. 2015, we are using a global climatological wind field based on European Remote-Sensing Satellite (ERS) satellite product and TAO observations (Menkes et al., 1998). We will better explain the origin of the different forcing fields in the next manuscript version.

**Bibliography:**

Aumont, O., Ethé, C., Tagliabue, A., Bopp, L., Gehlen, M.: PISCES-v2: an ocean biogeochemical model for carbon and ecosystem studies, Geosci Model Dev, 8, 2465–2513, doi:10.5194/gmd-8-2465-2015, 2015.

Bourgeois, T., Orr, J.C., Resplandy, L., Terhaar, J., Ethé, C., Gehlen, M., Bopp, L.: Coastal-ocean uptake of anthropogenic carbon, Biogeosciences, doi:10.5194/bg-13-4167-2016, 2016.

Menkes, C., Boulanger, J.-P., Busalacchi, A. J., J. Vialard, J., Delecluse, P., McPhaden, M. J., Hackert, E., and Grima, N.: Impact of TAO vs. ERS wind stresses onto simulations of the tropical Pacific Ocean during the 1993–1998 period by the OPA OGCM, in: Climatic Impact of Scale Interactions for the Tropical Ocean-Atmosphere System, EuroClivar Workshop Report, 46–48, 1998.

Zhang, Y., Xie, H., Fichot, C.G., Chen, G.: Dark production of carbon monoxide (CO) from dissolved organic matter in the St. Lawrence estuarine system: Implication for the global coastal and blue water CO budgets, J. Geophys. Res., 113, doi:10.1029/2008JC004811, 2008. Geophys. Res., 113, doi:10.1029/2008JC004811, 2008.

---

## Author Comment (AC2) · 30 Nov 2018

**Response for Referee #2**

We thank the referee for his comments and suggestions. We will strive to address each specific concern in detail.

**Minor comments:**

**Comment 1: Diurnal cycle and discretization choices**

If I get it correctly, for simulating CO you decided to choose a model setup that uses climatological forcing that does not resolve diurnal light cycle. Forcing data like NCEP or ERA interim however do resolve SW radiation fluxes on a 6 or 3 hourly basis. Given that you are using a sophisticated wave-length dependent CO production mechanism omitting on the other hand this feature seems like an odd decision on complexity versus simplification. As production and decay rates are highly uncertain, does resolving wave length dependency and omitting the day night cycle (thus also dark production, as described in Day and Faloona 2009, JGR Oceans 114) imply you expect the former to propagation stronger into the presented mean solution than the latter, i.e. more important to resolve in a realistic CO production scheme?

**Reply:** Indeed, forcing data like NCEP or ERA would enable us to resolve the diurnal cycle in terms of forcing fields. However, our choice for not including an explicit diurnal cycle for the CO photochemistry in our model set-up was mostly dictated by the consistency with the existing biogeochemical simulation (Aumont et al., 2015) in which we embed the CO module. In the standard version of NEMO-PISCES, as described in Aumont et al. (2015), the lack of an explicit diurnal cycle is justified by the fact that:

- the simple phytoplankton model that is used (constant stoechiometry for N and C) does not enable to explicitly represent the decoupling of the plankton internal carbon and nitrogen cycles that is a main feature of the plankton diurnal cycle (Flynn and Fasham, 2003).

- the vertical resolution in the upper part of the ocean (10 m for the first layers) does not enable to adequately represent the diurnal variability of the mixed layer depth.

We thought it would have been inconsistent to explicitly resolve the diurnal cycle for the photochemical part, but not for the biological part.

That said, because the photochemical production of CO is described, in our model, as a linear function of irradiance, the average photoproduction term over 24h is probably not very different from what we would have obtained from resolving the diurnal cycle.

These considerations will be added to the Discussion session. The potential inclusion of dark production is discussed in more details in the response to Referee 1. In brief, we plan to include the parameterization by Zhang et al. (2008) in our model set-up, to conduct some sensitivity experiments to be able to discuss the potential underestimation of CO production in our standard model set-up.

**Comment 2: Seasonal cycle**

How does the seasonal cycle of CO look like in the model? Is the quality of the model solution different for different seasons, i.e. how does it relate to the models' ability to represent the seasonal cycle of Chla?

**Reply:** We will add to the revised manuscript a brief description of the seasonal cycle of CO (figure and text).

For the evaluation of the Chla solution, we will recall in the discussion section the main conclusions from Aumont et al. (2015) and will include in the supplementary material a few plots comparing surface chlorophyll to estimates from available data-sets. Briefly, in Fig. 7 of Aumont et al. (2015), the simulated distribution is compared to GlobColour satellite observations for 2 seasons: Apr-May-Jun and Nov-Dec-Jan. The observed patterns are qualitatively reproduced by the model. Too low concentrations are simulated in the subtropical gyres, which is attributed to the lack of phytoplankton acclimation to oligotrophic conditions or to the assumption of constant phytoplankton stoichiometry. Chlorophyll concentrations are quite strongly underestimated in the equatorial Atlantic and in the Arabian Sea. In the latter region, mesoscale and submesoscale processes have been shown to be of critical importance. In two of the three main HNLC regions, i.e., the equatorial Pacific and the eastern subarctic Pacific, the model succeeds in reproducing the moderate chlorophyll concentrations. In the Southern Ocean, the third and largest of the principal HNLC regions, chlorophyll concentrations appear to be strongly overestimated by the model when evaluated against satellite-derived observational products, especially during summer. Furthermore, the increase in phytoplankton in late spring and early summer occurs too early.

**Comment 3: In situ data processing**

3.1. In the evaluation of the concentrations you are using model data collocated to observational data. Does this mean you are using individual grid cells? If so, are these representative for a larger surrounding area – did you consider averaging several grid cells, as physical features such as the extent of subtropical gyres, location of fronts etc. are not georeferenced, i.e. collocated with real world conditions during the ship cruises? In particular, for the vertical profiles it could be useful instead of showing one profile adding its variability taking into account several neighbouring cells (and eventually temporal standard deviations for within the averaging period).

**Reply:** The referee is right. We are using individual grid cells of the model output, which are co-located with the observations (this will be made explicit in the revised manuscript).

We did not consider averaging several grid cells as the model resolution is already very coarse (2°x2°). It would have been interesting to consider adjacent cells to capture specific physical features that are not exactly geo-referenced. However, this is limited first by the fact that we don't have access to the physical features associated to the in situ data and second by the fact that we are comparing model output which are climatological (with no inter-annual variability) with in situ data measured for a specific year and month.

3.2. Given that the compilation of observational data is presented to be unique and its averaging methods are very diverse I suggest to extend the section observations. E.g. it would be interesting to learn, also in light of the large temporal variability of CO concentrations, if certain months/seasons are better resolved in the observations than others. Now I can deduce this only from the tables, but do not get any direct information in the main text.

**Reply:** yes, we will better present and discuss the spatial and temporal coverage of the in situ data in the Materials section, instead of presenting it only in the tables.

3.3. It would be important to know on how many values per averaging period the temporal means are based, as e.g. the observed diurnal cycle is very strong and party not symmetric (afternoon maxima). Furthermore, I do not know if CO measurement techniques are comparable across the different observational sources wrt the limit of quantification / detection, or if there was development in the methodology from the 1970s to now. Are all of the published observational data equally reliable, in particular wrt to low CO values?

**Reply:** It is difficult to gather in situ data from literature as the quality of the metadata associated with measurements are very heterogeneous. For example, information about the time of the day at which the measure had been made is not always given, or in other cases the CO concentration given in the literature is already a mean upon a few measures taken during the day. Furthermore, it is true that the different data sets might not be equally reliable as the measurement's techniques have evolved since the late 70's, but this issue is not easily quantifiable. In the tables, we did our best to inform the reader about the origin and significance of the data used. We would like to emphasize that it is the first effort to gather observations from the first cruises in the 70' up to now in a CO observation-based data set.

**Comment 4: Choices of the manuscript structure –**

I struggled with the structure of the manuscript, presenting first simulation results and evaluation of the "standard" experiments followed by a separate discussion of the sensitivity experiments. Statements as "indicating again a possible bias in the production process" in the section on the standard experiment could be easily complemented by the results of the sensitivity studies on process parameterization, instead of having to collect this information later in the manuscript. Also this separation of the evaluation of surface concentrations leads to inconsistent level of discussion of potential sources of discrepancies: Whereas in the standard experiment it was argued that missing processes related to sea ice or a missing spatial variability of the decomposition rate might be causes of discrepancies in polar regions, only later in the section of CO production the authors state that also missing terrestrial CDOM sources might be a source of model data discrepancies. It is not clear to me how it was decided which of the parameterizations are chosen to be "standard" vs "alternative". For example, I understand that choosing the Launois et al. 2015 CDOM parameterization leads to high CO production and in combination with a consumption rate of 0.2 d-1 CO concentrations get too high compared to observations. On the other hand, using the "standard" parameterization together with a consumption rate of 1 d-1 lead to very low CO concentrations. The combination of the Launois et al. 2015 CDOM parameterization and a fast consumption rate was however not tested or presented without commenting on the reasons for this. If instead the authors would present all of the tested parameterizations in the methods part equally, present first an evaluation of the model results wrt to the range of parameters chosen and process parameterizations, and resulting from this discuss the source and sink budget, and emissions only for the most successful configuration, decisions taken and its consequences could become clearer.

**Reply:** We did initially chose a structure showing equally the different simulations with their respective evaluation. However, this structure was cumbersome as we had to evaluate against oceanic CO concentrations each simulation. Moreover, it did not permit to clearly highlight our "best guess" simulation (which we called "standard"), that we propose for use by the chemical atmospheric community. Indeed, considering the different tests performed and parameterizations, we think our standard simulation to be the best based on its CDOM parameterization. The section 'Sensitivity to alternative parameterizations' should be considered as a presentation of the possible range around the standard simulation. Simulations with a combination of the Launois et al. (2015) CDOM parameterization and a faster consumption rate, or with Preiswerk and Najjar (2000) with a slower consumption rate were indeed tested (but not included in the current version). Considering the present comment, we

plan to add and discuss these other tests in the next manuscript version.

**Comment 5: Lack in the physical model evaluation**

The model evaluation is lacking discussion of the simulated physical ocean solution, e.g. whereas contributions of MLD are mentioned in possible causes of discrepancies of modelled and observed CO concentrations these are not compared to the ship cruise or climatological T, S, or MLD data. I guess it would be possible to get the CTD data of the ship cruises and compare them. It would be useful to see in particular in the analysis of the vertical profiles, but also for the surface data whether how NEMO performs in regions with CO data.

**Reply:** As mentioned in reply for comment 2 for the Chla solution, in the supplementary material of the revised manuscript, we will include a few plots comparing SST and Mixed Layer Depth for different seasons to estimates from available data-sets. The dynamical state of NEMO used here is partly evaluated elsewhere: e.g., Mixed Layer Depth in Southern Ocean (Person et al., 2018), Water masses and transport (Iudicone et al., 2016), but no one publication offers an extensive evaluation of the physics used here. However, for the in situ data collected, it is difficult to gather the corresponding physical features as most of the time the CTD data of the ship cruises are not given along with the CO concentrations.

**Comment 6: Constant atmospheric CO**

The authors assume a constant homogeneous atmospheric mixing ratio of CO in their emission calculation. As major sources of CO (fossil fuel combustion, biomass burning) are on land and a major sink is reaction with of a large hemispheric and seasonal variation of CO mixing ratios in air due to the continental distribution and OH seasonality is expected. Is the over-saturation of the ocean indeed that strong that these variations can be omitted in the emission calculation?

**Reply:** Tests had previously been carried out on the atmospheric CO but were not shown in the submitted version of the manuscript. These tests show that the value of the atmospheric CO dry mole fraction is of little influence on the oceanic emission. For example, using a homogeneous and constant CO dry mole fraction of 45 pptv leads to a global oceanic CO emission of 3.7 Tg C yr$^{-1}$ (against 3.6 Tg C yr$^{-1}$ using 90 pptv). We will add these results in the next manuscript version.

**Comment 7: Clarification of a few points**

- p8 l23: ... same forcing fields as the ones in Aumont et al. 2015. Please help the reader to easily understand implications of your model setup into the results by repeating main

characteristics of that forcing (source, spatial and temporal resolution).

**Reply:** The main characteristics of the forcing will be better detailed in the next manuscript version.

- p10 l10: … all vertically integrated over the upper 1000 m. The vertical profiles suggest that below the euphotic zone there is not much CO left, why do you choose to integrate over 1000 m?

**Reply**: Yes, the integration upon 1000 m is nearly equivalent with an integration over the whole water column as almost no CO remains under that depth. This will be made clearer in the revised version.

- p12 l24 … which can be related to differences in the light penetration and mixed layer depth. Please be more specific… is the simulated MLD generally too low/high? Do you indicate a different mixing scheme would improve the profiles? How does the model's vertical resolution in the upper ocean affect the vertical profiles?

**Reply:** see reply to comment 5.

**Comment 8: CDOM comparisons**

P13 l17: … those minimums are best represented by the relation of Morel since the ones in Launois … give too high and too low values. Could you please comment on the simulated Chl field here, so that it gets clear that these CDOM parameterizations are responsible for the discrepancies rather than the simulated Chl. Furthermore, satellite derived observations are based on a number of assumptions (e.g. also wrt to light penetration depth in turbid and non-turbid waters) and models (bio-optical, atmospheric correction…), in particular in derived product as CDM. Furthermore, the discussion of the evaluation could be more detailed in discussion the quality of the Chla solution, which influences CO production.

**Reply:** we will better discuss the Chla evaluation as mentioned in the reply to comment 2. This will help to clarify the respective roles of CDOM parameterization versus Chla representation for the CDOM representation. In particular, we will complete Fig. 8 by adding a plot showing modelled and satellite-derived Chla as a function of latitude.

**Comment 9: Inter-annual variability**

Conclusions ii) […] the model does not consider the inter-annual variability of ocean physics and biogeochemistry […] Even with a climatological forcing both ocean physics and biogeochemistry solutions will show inter-annual variability due to e.g. fluid dynamics (wave propagation) and different plankton over-wintering stocks. Clearly this variability will be lower

than in simulations using transient forced boundary conditions or coupled atmosphere-ocean simulations. I suggest to rephrase this statement slightly.

**Reply:** We may not have been clear enough in the description, of the simulation protocol. In fact, we use an offline configuration of NEMO-PISCES model, i.e. PISCES is run using a climatological ocean dynamical state obtained from a previous NEMO physics-only simulation. We plan to better explain the experimental setup in the next manuscript version. In brief, the dynamical state of the ocean has been simulated with NEMO 3.2 (Madec, 2008) under climatological forcing as described in Aumont et al. (2015). The ocean model is first spun up for 200 years starting from the climatology of Conkright et al. (2002) for temperature and salinity. We then use the last year of these dynamical fields at 5-day temporal mean resolution (ocean currents, temperature, salinity, mixed layer depth, surface radiation…) to force the biogeochemical model, which is then spun up for 3000 years. We then turn on the CO module for an additional 2 years and analyse the last year of the simulation.

By construction, we do not simulate any inter-annual variability: the atmospheric forcing fields are climatological fields, the ocean dynamical state is also climatological as well as our biogeochemical simulation. In case of high spatial model resolution (> ¼°), it has been shown that an ocean-only simulation forced by climatological atmospheric forcing fields could indeed show some intrinsic variability at the inter-annual scale (Sérazin et al., 2018). This is clearly not the case at the typical resolution we're running our ocean-biogeochemical model.

**Comment 10:**

Fig.7: I find the colour coding of the lines confusing... is it time or location, what does it show in the subplot 'Swinnerton and Lamontagne 1973'?

**Reply:** Inside each subplot, one colour is for one profile. The dash line shows in situ measured profiles whereas full line shows the model profile co-located in space and time with the measured profile. This will be made clearer in the figure's legend.

**Bibliography:**

Aumont, O., Ethé, C., Tagliabue, A., Bopp, L., Gehlen, M.: PISCES-v2: an ocean biogeochemical model for carbon and ecosystem studies, Geosci Model Dev, 8, 2465–2513, doi:10.5194/gmd-8-2465-2015, 2015.

Conkright, M. E., R. A. Locarnini, H. E. Garcia, T. D. O'Brien, T. P. Boyer, C. Stephens, and J. Antonov, World Ocean Atlas 2001: Objective Analyses, Data Statistics and Figures, CD-ROM Documentation, Tech. rep., National Oceanographic Data Centre, Silver Spring, MD,

USA, 2002.

Flynn, K.J., and Fasham, M.J.R. (2003). Operation of light–dark cycles within simple ecosystem models of primary production and the consequences of using phytoplankton models with different abilities to assimilate N in darkness. J Plankton Res 25, 83–92.

Iudicone, D., K. B. Rodgers, Y. Plancherel, O. Aumont, T. Ito, R. M. Key, G. Madec, and M. Ishii, 2016: The formation of the ocean's anthropogenic carbon reservoir. Sci. Rep., **6**, 35473, doi: https://doi.org/10.1038/srep35473.

Launois, T., Belviso, S., Bopp, L., Fichot, C.G., Peylin, P.: A new model for the global biogeochemical cycle of carbonyl sulfide - Part 1: Assessment of direct marine emissions with an oceanic general circulation and biogeochemistry model, Atmos. Chem. Phys., 15, 2295–2312, doi:10.5194/acp-15-2295-2015, 2015.

Madec, G.: "NEMO Ocean Engine", Note du Pôle de Modélisation 27, Institut Pierre-Simon Laplace (IPSL), France, http://www.nemo-ocean.eu, 2008.

Person, R., Aumont, O., and Lévy, M. (2018). The Biological Pump and Seasonal Variability of pCO2 in the Southern Ocean: Exploring the Role of Diatom Adaptation to Low Iron. Journal of Geophysical Research: Oceans 123, 3204–3226.

Preiswerk, D. and Najjar, R.G.: A global, open-ocean model of carbonyl sulfide and its air-sea flux, Global Biogeochem Cycles, 14, 585–598, doi:10.1029/1999GB001210, 2000.

Sérazin, G., Penduff, T., Barnier, B., Molines, J.-M., Arbic, B.K., Müller, M., and Terray, L. (2018). Inverse Cascades of Kinetic Energy as a Source of Intrinsic Variability: A Global OGCM Study. J. Phys. Oceanogr. 48, 1385–1408.

---

## Author Response (AR1)

Dear Editor,

Following the suggestions of the reviewers, we have revised our manuscript and introduced a few major changes. The main ones are related to:
- the incorporation of a parameterization representing CO dark production into the standard version of the model.
- a more thorough discussion about the representation of the coastal zones in our global model.
- the addition of "new" in situ CO measurements in the data-base we use to evaluate the model simulations.
- a better description of the physical forcings used to force the biogeochemical model
- the addition of maps for the model evaluation (Chl-a, SST and MLD) in the Supplementary Material.

We thank the reviewers for their comments that have helped to improve our manuscript, and we hope these changes will meet your expectations

Best regards,

Ludivine Conte and co-authors

**Response for Referee #1**

**1) Major comments**

**Comment 1 — Coastal oceans** (including shelf areas, estuaries etc) are important sources of CDOM which, in turn, is the prerequisite of the photochemical production of CO. I am wondering why the role of coastal oceans is not discussed in the article. It is only mentioned briefly on page 17, lines 30-33. I understand that the model is not suitable to simulate coastal oceans (shelf areas, estuaries etc.). To this end, the authors should modify the ms title and the discussion by stating that their results are only valid for the open ocean or include a discussion of CO in coastal areas (i.e. contribution to CO emissions etc).

**Reply:** we agree that coastal areas could be important for the oceanic CO cycle.

In the revised manuscript version, we better discussed the implications of not resolving properly the coastal ocean in the discussion section.

In the Methods, P9, L20: « *Note that choosing a horizontal resolution of ~200km does not allow to fully resolve some fine-scale coastal processes such as tides or mesoscale and sub-mesoscale eddies and associated upwelling occurring in the costal ocean and hence these areas are represented with large uncertainties in the simulations performed.* »

In the Conclusion section, P19 L18: "*Regarding the spatial scale, our simulated concentrations are probably impacted by the very coarse resolution of the model. In particular, this is critical to resolve the coastal ocean, where a number of in-situ measurements of CO concentrations have been performed. Indeed, the ~100-to-200 km horizontal resolution does not allow to fully resolve some fine-scale coastal processes and hence our estimation of the oceanic CO cycle is rather suitable to the blue waters, although a crude representation of specific processes occurring in coastal areas exists (like riverine nutrients inputs, iron input by sediment re-suspension, or coastal upwellings).* »

**Comment 2 — Important literature has been ignored:**
- Kawagucci, S., et al. (2014). "Molecular hydrogen and carbon monoxide in seawater in an area adjacent to Kuroshio and Honshu Island in Japan." Mar. Chem. 164: 75-83.

**Reply:** In the revised manuscript version, we have considered the vertical CO profiles as well as the surface concentrations measured by Kawagucci et al.

- Park, K. and T. S. Rhee (2016). "Oceanic source strength of carbon monoxide on the basis of

basin-wide observations in the Atlantic." Environmental Science-Processes & Impacts 18(1): 104-114.

**Reply:** In the revised manuscript version, we have considered the oceanic CO concentrations measured by Park and Rhee, as well as discussed our global CO emission estimate against their estimate (4-24 Tg CO yr$^{-1}$).

In the Discussion, P18, L7: "*From Atlantic data, Stubbins et al. (2006a) estimated a yearly flux to the atmosphere of 3.7 ± 2.6 Tg C yr$^{-1}$ and Park and Rhee (2016) estimated the emissions in the range 1-12 Tg C yr$^{-1}$. From Pacific data, Zafiriou et al. (2003) estimated a flux of 6 Tg C yr$^{-1}$ and Bates et al. (1995) a range of 3-11 Tg C yr$^{-1}$.*"

- Xie, H. X. and O. C. Zafiriou (2009). "Evidence for significant photochemical production of carbon monoxide by particles in coastal and oligotrophic marine waters." Geophys. Res. Lett. 36.

**Reply:** In the revised manuscript version, we have discussed the potential CO photoproduction by organic particles

In the Conclusion section, P20 L9: "*Beyond a better knowledge of the CDOM pool, it must be mentioned that the whole organic matter pool needs to be better understood in order to better constrain photoproduction. Indeed, it has been suggested that even particulate organic matter could be a substrate for CO photoproduction (Xie and Zafiriou, 2009).*"

- Yang, G. P., et al. (2010). "Distribution, flux and biological consumption of carbon monoxide in the Southern Yellow Sea and the East China Sea." Mar. Chem. 122(1-4): 74-82.

**Reply:** we did consider Yang et al. oceanic CO measurements. However, due to the coarse model resolution and the very coastal location of the data, we are not able to include these data in our evaluation as there is no model grid cell associated to the location of their measurements.

**Comment 3: The dark production** (DP), which was shown by Zhang et al. (2008) to be a significant additional source of CO, has been ignored in the model approach (see equation (1)). However, in the conclusions (page 18, line 12-18) it is stated that '[...] analyses of the collected vertical profiles did not seem to clearly support the importance of such a mechanism to explain the differences with our simulated profiles.' This is too vague and not acceptable. I think that the correct scientific approach to tackle this 'problem' would be to include the DP (I guess you can use the parameterization given by Zhang et al., 2008) in equation (1) and show the results of model

runs with DP/without DP. Only based on these model results you will be able to assess the role of the DP.

**Reply:** The dark production process has been added to PISCES as described by Zhang et al. 2008. Resulting in a global dark production budget of 8.5 Tg C yr$^{-1}$, we chose to integrate the process in our standard model version. Hence, all previous estimates and figures had been changed consequently.

**2) Minor comments:**

**Comment 1:** Page 3, line 21: please give the correct chemical formulas for nitrate, ammonium, phosphate, and iron.

**Reply:** $NO_3$, $NH_4$, $PO_4$, Si and Fe, has been changed for $NO_3^-$, $NH_4^+$, $PO_4^{3-}$, $Si(OH)_4$ and dissolved Fe.

**Comment 2:** Page 6, section 2.1.4: please note that fCO is a '(dry) mole fraction' (it is not correct to call it a 'mixing ratio' or a 'concentration').

**Reply:** We have replaced the term 'mixing ratio' by 'dry mole fraction'.

**Comment 3:** Page 6, line 19: In view of the pronounced spatial and temporal variability of atm CO I am wondering why the atm CO was set to fixed global mean. Please discuss.

**Reply:** Tests had previously been performed about the atmospheric CO but were not shown in the manuscript version. They show that the value of the atmospheric CO dry mole fraction is of little influence on the oceanic emission. For example, using a homogeneous and constant CO dry mole fraction of 45 pptv leads to a global oceanic CO emission of 3.7 Tg C yr$^{-1}$ (against 3.6 Tg C yr$^{-1}$ using 90 pptv). We will specify these results in the next manuscript version.

In the Methods, P7, L13: *"Note that the global oceanic CO emissions have a low sensitivity to the atmospheric mole fraction. Hence, using a constant dry atmospheric mole fraction of 45 ppbv instead of 90 ppbv changes the global oceanic CO emissions by only 3%"*.

**Comment 4:** Page 7, wind speed: Please state whether you used a global mean wind speed (which value? ref?) or whether a global wind field (ref?) was used for the computation of the air/sea gas exchange.

**Reply:** As mentioned in Aumont et al. 2015, we are using a global climatological wind field based on European Remote-Sensing Satellite (ERS) satellite product and TAO observations (Menkes et al., 1998). We better explained the origin of the different forcing fields in the revised manuscript version.

See changes P9 L5, section 2.3 - Standard Experiment

had been changed consequently.

**Comment 2: Seasonal cycle**

How does the seasonal cycle of CO look like in the model? Is the quality of the model solution different for different seasons, i.e. how does it relate to the models' ability to represent the seasonal cycle of Chla?

**Reply:**

We have added to the revised manuscript a brief description of the seasonal cycle of CO.

In section 3.1.2 P11 L23: "*Figure 5 (panels B and D) also presents the seasonal variability of the surface patterns with latitude. Both CO and photoproduction surface patterns present a strong seasonal cycle with maximums reached at the end of spring and summer in both hemispheres »*

For the evaluation of the Chla solution, we have included in the supplementary material a few plots comparing surface chlorophyll to estimates from available data-sets.

**Comment 3: In situ data processing**

3.1. In the evaluation of the concentrations you are using model data collocated to observational data. Does this mean you are using individual grid cells? If so, are these representative for a larger surrounding area – did you consider averaging several grid cells, as physical features such as the extent of subtropical gyres, location of fronts etc. are not geo-referenced, i.e. collocated with real world conditions during the ship cruises? In particular, for the vertical profiles it could be useful instead of showing one profile adding its variability taking into account several neighbouring cells (and eventually temporal standard deviations for within the averaging period).

**Reply:** The referee is right. We are using individual grid cells of the model output, which are co-located with the observations (this will be made explicit in the revised manuscript).

We did not consider averaging several grid cells as the model resolution is already very coarse (2°x2°). It would have been interesting to consider adjacent cells to capture specific physical features that are not exactly geo-referenced. However, this is limited first by the fact that we don't have access to the physical features associated to the in situ data and second by the fact that we are comparing model output which are climatological (with no inter-annual variability) with in situ data measured for a specific year and month.

3.2. Given that the compilation of observational data is presented to be unique and its averaging methods are very diverse I suggest to extend the section observations. E.g. it would be interesting to learn, also in light of the large temporal variability of CO concentrations, if certain months/seasons are better resolved in the observations than others. Now I can deduce this only from the tables, but do not get any direct information in the main text.

**Reply:** We have added in the Materials section a sentence to better present the spatio-temporal coverage of the data.

In section 2.4 Comparison to in situ data, P9 L26: "*This dataset covers all seasons and latitudes from 80°N to more than 70°S. However, whereas the Atlantic and Pacific Ocean are fairly well covered, some wide areas are not documented (such as in the Indian ocean).*"

However, considering the diversity of the seasons and location covered it is difficult to better present it in the text. Details are in the tables 1 and 2.

3.3. It would be important to know on how many values per averaging period the temporal means are based, as e.g. the observed diurnal cycle is very strong and party not symmetric (afternoon maxima). Furthermore, I do not know if CO measurement techniques are comparable across the different observational sources wrt the limit of quantification / detection, or if there was development in the methodology from the 1970s to now. Are all of the published observational data equally reliable, in particular wrt to low CO values?

**Reply:** It is difficult to gather in situ data from literature as the quality of the metadata associated with measurements are very heterogeneous. For example, information about the time of the day at which the measure had been made is not always given, or in other cases the CO concentration given in the literature is already a mean upon a few measures taken during the day. Furthermore, it is true that the different data sets might not be equally reliable as the measurement's techniques have evolved since the late 70's, but this issue is not easily quantifiable. In the tables, we did our best to inform the reader about the origin and significance of the data used. We would like to emphasize that it is the first effort to gather observations from the first cruises in the 70' up to now in a CO observation-based data set.

**Comment 4: Choices of the manuscript structure –**

I struggled with the structure of the manuscript, presenting first simulation results and evaluation of the "standard" experiments followed by a separate discussion of the sensitivity experiments. Statements as "indicating again a possible bias in the production process" in the section on the standard experiment could be easily complemented by the results of the

sensitivity studies on process parameterization, instead of having to collect this information later in the manuscript. Also this separation of the evaluation of surface concentrations leads to inconsistent level of discussion of potential sources of discrepancies: Whereas in the standard experiment it was argued that missing processes related to sea ice or a missing spatial variability of the decomposition rate might be causes of discrepancies in polar regions, only later in the section of CO production the authors state that also missing terrestrial CDOM sources might be a source of model data discrepancies. It is not clear to me how it was decided which of the parameterizations are chosen to be "standard" vs "alternative". For example, I understand that choosing the Launois et al. 2015 CDOM parameterization leads to high CO production and in combination with a consumption rate of 0.2 d-1 CO concentrations get too high compared to observations. On the other hand, using the "standard" parameterization together with a consumption rate of 1 d-1 lead to very low CO concentrations. The combination of the Launois et al. 2015 CDOM parameterization and a fast consumption rate was however not tested or presented without commenting on the reasons for this. If instead the authors would present all of the tested parameterizations in the methods part equally, present first an evaluation of the model results wrt to the range of parameters chosen and process parameterizations, and resulting from this discuss the source and sink budget, and emissions only for the most successful configuration, decisions taken and its consequences could become clearer.

**Reply:** We did initially chose a structure showing equally the different simulations with their respective evaluation. However, this structure was cumbersome as we had to evaluate against oceanic CO concentrations each simulation. Moreover, it did not permit to clearly highlight our "best guess" simulation (which we called "standard"), that we propose for use by the chemical atmospheric community. Indeed, considering the different tests performed and parameterizations, we think our standard simulation to be the best based on its CDOM parameterization. The section 'Sensitivity to alternative parameterizations' should be considered as a presentation of the possible range around the standard simulation. Simulations with a combination of the Launois et al. (2015) CDOM parameterization and a faster consumption rate, or with Preiswerk and Najjar (2000) with a slower consumption rate were indeed tested (but not included in the current version).

Considering the present comment, we have discussed about these other tests in the revised manuscript version:

At the end of section 3.3 Sensitivity to alternative parameterizations, P17 L12: "*Based on these different tests, we show that the main terms controlling oceanic CO concentrations are still largely under-constrained, although available in situ CO concentrations, global budget estimates as well as the CDM concentrations retrieved from space are helping to choose a*

*standard, i.e. 'best guess' simulation. Here, the best parameterization of CDOM absorption compared to satellite products is given by Morel (2009), which then dictates our choice for the standard simulation. However, it is worth mentioning that using the parameterization of Launois et al. (2015) with a higher bacterial consumption rate, or the one of Preiswerk and Najjar (2000) with lower consumption rate would result in oceanic CO concentrations that give similar skill scores when compared to available CO concentration observations (tests not shown), but for which we have a less confidence in the CDOM parameterization.*

**Comment 5: Lack in the physical model evaluation**

The model evaluation is lacking discussion of the simulated physical ocean solution, e.g. whereas contributions of MLD are mentioned in possible causes of discrepancies of modelled and observed CO concentrations these are not compared to the ship cruise or climatological T, S, or MLD data. I guess it would be possible to get the CTD data of the ship cruises and compare them. It would be useful to see in particular in the analysis of the vertical profiles, but also for the surface data whether how NEMO performs in regions with CO data.

**Reply:** In the supplementary material of the revised manuscript, we have included a few plots comparing SST and Mixed Layer Depth for different seasons to estimates from available data-sets. The dynamical state of NEMO used here is partly evaluated elsewhere: e.g., Mixed Layer Depth in Southern Ocean (Person et al., 2018), Water masses and transport (Iudicone et al., 2016), but no one publication offers an extensive evaluation of the physics used here. However, for the in situ data collected, it is difficult to gather the corresponding physical features as most of the time the CTD data of the ship cruises are not given along with the CO concentrations.

**Comment 6: Constant atmospheric CO**

The authors assume a constant homogeneous atmospheric mixing ratio of CO in their emission calculation. As major sources of CO (fossil fuel combustion, biomass burning) are on land and a major sink is reaction with of a large hemispheric and seasonal variation of CO mixing ratios in air due to the continental distribution and OH seasonality is expected. Is the over-saturation of the ocean indeed that strong that these variations can be omitted in the emission calculation?

**Reply:** Tests had previously been carried out on the atmospheric CO but were not shown in the submitted version of the manuscript. These tests show that the value of the atmospheric CO dry mole fraction is of little influence on the oceanic emission. For example, using a homogeneous and constant CO dry mole fraction of 45 pptv leads to a global oceanic CO emission of 3.7 Tg C yr$^{-1}$ (against 3.6 Tg C yr$^{-1}$ using 90 pptv).

In the Methods, P7, L13: "*Note that the global oceanic CO emissions have a low sensitivity to the atmospheric mole fraction. Hence, using a constant dry atmospheric mole fraction of 45*

*ppbv instead of 90 ppbv changes the global oceanic CO emissions by only 3%".*

**Comment 7: Clarification of a few points**

- p8 l23: ... same forcing fields as the ones in Aumont et al. 2015. Please help the reader to easily understand implications of your model setup into the results by repeating main characteristics of that forcing (source, spatial and temporal resolution).

**Reply:** We have clarified the section 2.3 Standard experiment.

- p10 l10: … all vertically integrated over the upper 1000 m. The vertical profiles suggest that below the euphotic zone there is not much CO left, why do you choose to integrate over 1000 m?

**Reply**: Yes, the integration upon 1000 m is nearly equivalent with an integration over the whole water column as almost no CO remains under that depth.

  - p12 l24 … which can be related to differences in the light penetration and mixed layer depth. Please be more specific… is the simulated MLD generally too low/high? Do you indicate a different mixing scheme would improve the profiles? How does the model's vertical resolution in the upper ocean affect the vertical profiles?

**Reply:** see reply to comment 5.

**Comment 8: CDOM comparisons**

P13 l17: … those minimums are best represented by the relation of Morel since the ones in Launois … give too high and too low values. Could you please comment on the simulated Chl field here, so that it gets clear that these CDOM parameterizations are responsible for the discrepancies rather than the simulated Chl. Furthermore, satellite derived observations are based on a number of assumptions (e.g. also wrt to light penetration depth in turbid and non-turbid waters) and models (bio-optical, atmospheric correction…), in particular in derived product as CDM. Furthermore, the discussion of the evaluation could be more detailed in discussion the quality of the Chla solution, which influences CO production.

**Reply:** we have completed Fig. 8 by adding a plot showing modelled and satellite-derived Chla as a function of latitude. Also, in the text section 3.3.1 CDOM parameterization, we have better explained the CDOM solution regarding the Chla solution.

**Comment 9: Inter-annual variability**

Conclusions ii) […] the model does not consider the inter-annual variability of ocean physics and biogeochemistry […] Even with a climatological forcing both ocean physics and

biogeochemistry solutions will show inter-annual variability due to e.g. fluid dynamics (wave propagation) and different plankton over-wintering stocks. Clearly this variability will be lower than in simulations using transient forced boundary conditions or coupled atmosphere-ocean simulations. I suggest to rephrase this statement slightly.

**Reply:** We have clarified the section 2.3 Standard experiment. By construction, we do not simulate any inter-annual variability: the atmospheric forcing fields are climatological fields, the ocean dynamical state is also climatological as well as our biogeochemical simulation. In case of high spatial model resolution (> ¼°), it has been shown that an ocean-only simulation forced by climatological atmospheric forcing fields could indeed show some intrinsic variability at the inter-annual scale (Sérazin et al., 2018). This is clearly not the case at the typical resolution we're running our ocean-biogeochemical model.

**Comment 10:**

Fig.7: I find the colour coding of the lines confusing... is it time or location, what does it show in the subplot 'Swinnerton and Lamontagne 1973'?

**Reply:** Inside each subplot, one colour is for one profile. The dash line shows in situ measured profiles whereas full line shows the model profile co-located in space and time with the measured profile.

[revised manuscript text omitted]

Éviter veuves et orphelines, Ajuster l'espace entre le texte latin et asiatique, Ajuster l'espace entre le texte asiatique et les nombres, Tabulations :Pas à  0,99 cm +  1,98 cm +  2,96 cm +  3,95 cm +  4,94 cm +  5,93 cm +  6,91 cm +  7,9 cm +  8,89 cm +

| Page 29 : [3] Supprimé | Utilisateur de Microsoft Office | 04/01/2019 17:35 |

| Page 32 : [4] Mis en forme | Utilisateur de Microsoft Office | 04/01/2019 17:35 |

Éviter lignes veuves et orphelines, Ne pas ajuster l'espace entre le texte latin et asiatique, Ne pas ajuster l'espace entre le texte et les nombres asiatiques, Tabulations : 0,99 cm, À gauche +  1,98 cm, À gauche +  2,96 cm, À gauche +  3,95 cm, À gauche

| Page 33 : [5] Supprimé | Utilisateur de Microsoft Office | 04/01/2019 17:35 |